



# Evaluation of WRF-Chem simulated meteorology and aerosols over northern India during the severe pollution episode of 2016

Prerita Agarwal[1], David S. Stevenson[1], Mathew R Heal[2]

[1]School of GeoSciences, University of Edinburgh, Crew Building, Edinburgh, EH9 3FF, UK,

[2]School of Chemistry, University of Edinburgh, Joseph Black Building, Edinburgh, EH9 3FJ, UK

*Correspondence to*: Prerita Agarwal (prerita.agarwal@ed.ac.uk), David Stevenson
(david.s.stevenson@ed.ac.uk)

## Abstract

Severe seasonal air pollution events have become frequent over northern India, particularly over the
Indo-Gangetic Plain (IGP). These episodic hazes, marked by exceedingly high levels of ambient $PM_{2.5}$
(particulate matter having an aerodynamic diameter $\leq$ 2.5 microns), are hazardous for visibility and
public health. It is therefore imperative to examine the capabilities of current state-of-the-art coupled
meteorology-chemistry models at predicting air quality over this region. We provide a comprehensive

evaluation of WRF-Chem (v4.2.1) simulated seasonal meteorology and aerosol chemistry ($PM_{2.5}$ and
its black carbon (BC) component) using a range of ground-based, satellite and reanalysis products, with
a focus on the November 2016 haze episode. Daily and diurnal features in simulated 2 m temperature
show best agreement followed by relative humidity with overall low biases. Upper air meteorology
comparisons with radiosonde observations show excellent model skill in reproducing the vertical

temperature gradient ($r > 0.95$). Both ground and radiosonde observations confirm systematic
overestimations in simulated surface wind speeds (by $\sim 0.5 - 0.8$ m s$^{-1}$), driven by high nocturnal biases.
Modelled $PM_{2.5}$ concentrations generally compare well with the ground-based measurements in
October-November (post-monsoon) but are strongly overestimated (by a factor of 2) in September
(monsoon) due to dust constituent. Delhi experiences some of the highest daily mean $PM_{2.5}$

concentrations within the study region with largest biases during the extreme pollution episode.
Dominant anthropogenic components in the modelled $PM_{2.5}$ in Delhi during the episode include nitrate
($\sim 25$ %), followed by secondary organic aerosols ($\sim 20$ %), and primary organic matter, and elevated
BC concentrations. Modelled spatiotemporal $PM_{2.5}$ and BC compare well with MERRA-2 products.
Spatially, high aerosol optical depth (AOD) over the IGP is accurately represented by the model relative

to MODIS satellite ($r \geq 0.8$), and ground-based AERONET observations ($r \geq 0.69$), except during
September. Generally, WRF-Chem correctly represents the meteorology during the afternoon and has
a reasonable ability to reproduce wind patterns. This (among other factors like imperfect representation
of emissions and land use information) plays a key role in dust overestimations in monsoon and
anthropogenic aerosol underestimations in post-monsoon owing to enhanced dilution and mixing in the

model. Overall, we find the model suitable to understand the aerosol feedbacks on meteorology during
extreme pollution events with an improved diurnal characterisation of boundary layer processes and
emissions estimates.



## 1. Introduction

Atmospheric particle pollution in India is a persistent environmental issue and a leading health risk
factor for its 1.4 billion population (Pandey et al., 2021). In 2019, ambient air pollution was estimated
to cause almost a million premature deaths in India (Pandey et al., 2021). The State of Global Air 2022
(HEI, 2022) reports that over 90% of the Indian population resides in areas where the annual mean
concentrations of $PM_{2.5}$ (particulate matter having an aerodynamic diameter smaller than 2.5 microns)
exceed even the minimal interim target of 35 µg m$^{-3}$ recommended by the World Health Organization
Air Quality Guidelines (WHO 2021). The country is home to 18 of the 20 cities worldwide with the
greatest rise in $PM_{2.5}$ pollution in the last decade (HEI, 2022). This upward trend in degraded air quality
is projected to continue across South Asia under current policies, including more frequent high pollution
incidents over northern India (Kumar et al., 2018c; Paulot et al., 2022). These trends have huge
consequences for the future life expectancy of the 400 million residents of this region which is currently
reported to be reduced by more than 9 years under the current pollution burden (Greenstone and Fan|,
2020).

The Indo-Gangetic Plain (IGP) is situated south of the Himalayas and stretches from parts of Pakistan
in the west, through north and east India and Nepal, to Bangladesh in the east. The IGP is a heavily
populated region (home to over 40% of the total Indian population) with a large number of rural,
suburban and urban clusters (Fig. 1a). It is characterised by intensive multi-cropping systems, rapid
industrialisation and a growing economy, which results in a heterogeneous mix of particle and gaseous
pollutant emissions (Venkataraman et al., 2018; Kumar et al., 2020a). The region is a global centre for
poor air quality (Singh et al., 2017), underpinned by India being one of the largest emitters of
anthropogenic aerosols in the world (Lu et al., 2011). The anthropogenic sources include vehicles,
industry, burning of crop-waste and garbage, residential cooking and mining. The emissions
contributions are dominantly composed of sulfate precursors and carbonaceous aerosols, driven by a
rapid increase in demand for energy (Lu et al., 2011). Black carbon (BC) is fine particulate matter's
light-absorbing component (Lack and Cappa, 2010) and is released during incomplete combustion of
carbon-containing fossil fuels like coal, oil and gas, and biofuels like wood, agricultural residues and
forest fires. BC particles are short-term climate forcers with a net positive radiative forcing
(Ramanathan et al., 2001; Bond et al., 2013; Wang et al., 2014). BC emissions from India are one of
the highest globally and significantly impact the Indian summer monsoon, regional climate, and human
health (Ramanathan et al., 2001). Natural particle sources such as mineral dust also substantially
influence the air quality over the IGP and broader northern India (Li et al., 2017). Additionally, air
quality over the IGP region is greatly affected by the prevailing meteorology, topography and the long-
range transport of pollutants (Kaskaoutis et al., 2014; Kumar et al., 2014; Schnell et al., 2018; Ojha et
al., 2020).



In addition to the year-round poor air quality over the IGP region, recurring intense post-monsoon and winter haze episodes have been reported in numerous studies (Ram et al., 2016; Kanawade et al., 2020; Beig et al., 2019; Bharali et al., 2019; Thomas et al., 2019; Dhaka et al., 2020; Kumar et al., 2018a; Kumari et al., 2021; Gupta et al., 2022). Most of these severe episodes coincide with the biomass burning period (mid-October to November) during which agricultural land is cleared by burning crop residues, primarily paddy, in open fields (Singh et al., 2020). Although highly seasonal, the emissions from these multiple small to large fires emit large amounts of reactive gases and particles such as carbon monoxide (CO), nitrogen oxides ($NO_X$), volatile organic compounds (VOCs), carbonaceous particles and other components of $PM_{2.5}$ and $PM_{10}$ (Singh et al., 2020; Kumar et al., 2021). One such severe haze event over northern India occurred between the end of October and 7[th] November 2016 leading to daily mean $PM_{2.5}$ concentrations of 300-600 µg m$^{-3}$, some 20-40 times greater than the 24-h WHO 2021 Air Quality Guideline of 15 µg m$^{-3}$ (Mukherjee et al., 2018; Sawlani et al., 2019; Kanawade et al., 2020; Jethva et al., 2019). Jethva et al (2019) reported that crop-residue fire counts over northwest India were particularly high in the 2016 post-monsoon period. Alongside crop biomass burning emissions, the unfavourable meteorology and accumulation of local urban emissions contributed to this week-long extremely high pollution episode (Kanawade et al., 2020; Sawlani et al., 2019).

Modelling studies characterising air pollution over India have utilised a variety of regional chemistry transport models (Nair et al., 2012; Kumar et al., 2012a, b; Moorthy et al., 2013; Pan et al., 2015; Srivastava et al., 2016; Schnell et al., 2018; Ghosh et al., 2023). These studies highlight various problems in simulating atmospheric composition over the Indian subcontinent such as capturing the high aerosol loading, erroneous boundary-layer parametrizations, underestimations in emissions inventories, complex mountain topography and inaccurate moisture transport. This is especially true for simulations of surface BC concentrations, which utilise regional South Asian emissions inventories that are thought to underestimate the BC emissions (Kumar et al., 2015; Govardhan et al., 2019). Equally important is simulating the vertical distribution of BC particles and understanding their effect on atmospheric stability, for which only limited measurements have been made over India. These studies find high BC loadings vertically (up to 8 km altitude) over northwest and central India during different months (Babu et al., 2011; Bisht et al., 2016; Brooks et al., 2019). The role of these absorbing particles in modifying the vertical boundary layer structure during a haze episode over the northern Indian region has been poorly explored to date (Bharali et al., 2019).

This study aims to evaluate the WRF-Chem regional atmospheric chemistry transport model for understanding the spatiotemporal aerosol-planetary boundary layer dynamics across north India and the IGP in September-November 2016. First, WRF-Chem simulations of surface and vertical meteorology are evaluated against multiple available observations from weather stations and radiosonde profiles and reanalysis datasets. Secondly, the modelled chemistry and aerosol optical properties are evaluated against reanalysis products, satellite, and ground-based measurements. The study focuses on



characterising monsoon to post-monsoon changes in meteorology and the atmospheric chemical

composition of PM$_{2.5}$ and BC.

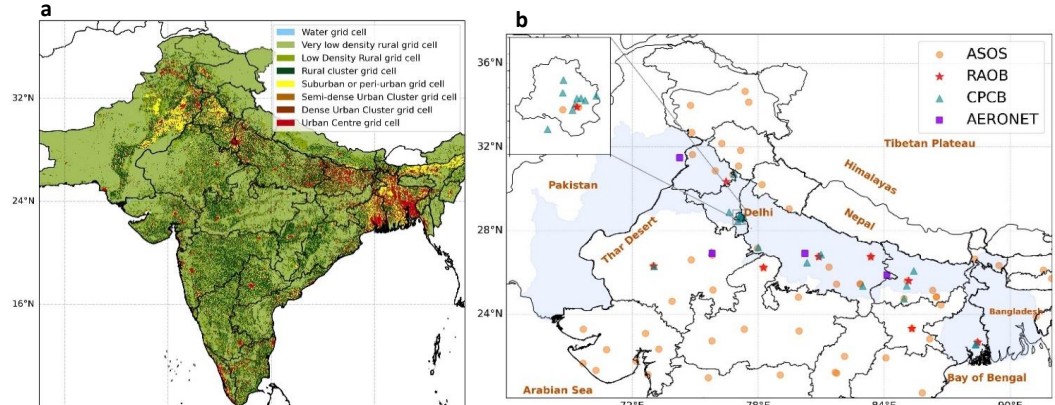

**Figure 1**. a) Degree of urbanisation based on 2015 human population size and built-up area density data over India from GHS-SMOD (Schiavina et al., 2023) b) Locations of the observation sites used for comparison in this study; the legend indicates the different datasets (ASOS: automatic weather stations, RAOB: Radiosonde observations, CPCB: Indian Central Pollution Control Board PM$_{2.5}$ ground monitoring stations, AERONET: Aerosol Robotic Network ground remote sensing observations). The inset figure is an enlarged map of Delhi capital and the geographical area falling under IGP region is highlighted in light blue colour.

## 2. Data and Methods

### 2.1 WRF-Chem model description and configuration

The Weather Research and Forecasting model (version 4.2.1) coupled with Chemistry (WRF-Chem)

(Grell et al., 2005; Fast et al., 2006) is an atmospheric chemistry transport model widely applied to the South Asia region, including its development as an air quality early-warning system for Delhi (Jena et al., 2021; Kumar et al., 2020b). It has a terrain-following vertical coordinate system and is available with a range of physical parameterizations (Skamarock et al., 2008). The transport of trace gases and aerosol species in WRF-Chem uses identical vertical and horizontal coordinates, allowing for feedbacks

between meteorology and chemistry via radiation and photolysis (Grell et al., 2005). This makes WRF-Chem well-suited for investigating and isolating the interactions between aerosols and meteorology.

The single domain for this study covers the northern part of South Asia (20 – 38° N and 66 – 90° E) at 12 km horizontal resolution (Fig. 1b), with 33 vertical levels from the surface to the model top which is fixed at 50 hPa. The lowest 10 levels are below 1 km. The configurations of WRF-Chem dynamical



and chemical parametrizations used in this study are adopted from the literature available for South
     Asia and are summarized in Supplementary Table S1. Hourly fifth-generation European Centre for
     Medium-Range Weather Forecasts (ECMWF) reanalysis (ERA5) data at a horizontal resolution of
     $0.25° \times 0.25°$ is used for initializing the meteorology, boundary conditions and nudging in the model
     (Hersbach et al., 2020). The WRF model temperature, horizontal winds and water vapour are nudged
(relaxation time of $6 \times 10^{-4}$ s$^{-1}$) to ERA5 value. The input static terrestrial and land-use data are obtained
     from MODIS IGBP 21-category land-cover classification (Friedl et al., 2002).

     The time-varying boundary conditions for chemistry are taken from the global 6-hourly Model for
     Ozone and Related Chemical Tracers (MOZART-4)/ Goddard Earth Observing System Model version
     5 (GEOS, National Center for Atmospheric Research 2016). The simulation of gas-phase chemistry in
WRF-Chem is provided by the updated MOZART-4 (Emmons et al., 2010, 2020) scheme which
     includes treatment of biogenic hydrocarbons and aromatics (Hodzic and Jimenez, 2011; Knote et al.,
     2014). Description of aerosol chemistry, including organic aerosols, is provided by the Model for
     Simulating Aerosol Interactions and Chemistry (MOSAIC) 4-bin scheme (Zaveri et al., 2008). The
     MOSAIC scheme uses a sectional approach to divide dry aerosol diameter into four discrete bins:
0.039–0.156 µm, 0.156–0.625 µm, 0.625–2.5 µm and 2.5–10 µm (the coarse PM bin) (Zaveri et al.,
     2008). The aerosol distribution scheme includes both in-cloud and impaction scavenging and assumes
     aerosols to be internally mixed within the same bin and externally mixed between the bins (Riemer et
     al., 2019). MOSAIC simulates sulfate ($SO_4^{2-}$), nitrate ($NO_3^-$), ammonium ($NH_4^+$), calcium ($Ca^{2+}$),
     carbonate ($CO_3^{2-}$), black carbon (BC), primary organic mass (OM), liquid water ($H_2O$), sea salt (NaCl)
and other inorganic species such as minerals and trace metals (Zaveri et al., 2008). The Fast
     TrophosphericUltraviolet–Visible (FTUV) photolysis scheme (Tie et al., 2003) provides photolysis rates
     and accounts for the aerosol feedback on photolysis (Hodzic and Knote, 2014).

     Monthly anthropogenic emissions at $0.1° \times 0.1°$ horizontal resolution are obtained from 2010 EDGAR-
     HTAPv2.2 (Emission Database for Global Atmospheric Research for Hemispheric Transport of Air
Pollution version 2.2, https://edgar.jrc.ec.europa.eu/dataset_htap_v2). The emission sectors included in
     EDGAR-HTAPv2.2 are industrial, residential, transportation, agriculture, shipping, energy, and
     aviation. For emissions from India, EDGAR-HTAPv2.2 incorporates the regional emissions inventory
     from the Model Inter-comparison Study for Asia Phase III (MICS-Asia III) to derive emissions maps
     at a common grid resolution of $0.1° \times 0.1°$ (Janssens-Maenhout et al., 2015). Under MICS-ASIA, a
mosaic of regional anthropogenic emission inventories (MIX) was developed by combining the
     nationally reported estimates by Argonne National Laboratory (ANL-India) and REAS2 (Regional
     Emission inventory in Asia) (Lu et al., 2011; Li et al., 2017). The total emissions for $SO_2$, $NO_x$, $NH_3$,
     $PM_{10}$, $PM_{2.5}$, BC, OC and non-methane volatile organic compounds (NMVOCs) are speciated in the
     model following the MOSAIC-MOZART chemistry mechanism. The anthropogenic emissions input
have a simplified diurnal variation implemented where two sets of diurnal files are generated with each



file having identical 12 hourly values for all the pollutants. The use of EDGAR-HTAPv2.2 inventory from 2010 estimates adds some uncertainties to the model results. However, compared to other global inventories (e.g., ECLIPSE) of coarser resolution, the use of EDGAR-HTAPv2.2 has been found to simulate air quality over India satisfactorily (Saikawa et al., 2020; Upadhyay et al., 2020).

In India, the post-harvest agricultural residue is largely cleared by burning it in open fields, and this is a dominant contributor to Indian $PM_{2.5}$, BC, OC, $SO_2$, $NO_x$ and NMVOC emissions (Venkataraman et al., 2018). As EDGAR emissions do not include any biomass burning emissions (from agricultural fires, wildfires or prescribed fires), these are derived from the Fire Inventory from NCAR, version 1.5 (FINNv1.5) (Wiedinmyer et al., 2011). The emissions are based on satellite-measured locations of

active fires and emission factors relevant to the underlying land cover (Akagi et al., 2011). The FINNv1.5 fire emissions inputs are distributed at 1 km spatial and hourly temporal resolution for 2016 (https://www.acom.ucar.edu/Data/fire/).

Biogenic emissions are calculated online (updated every 30 minutes) using the Model of Emissions of Gases and Aerosol from Nature (MEGAN v2.0) (Guenther et al., 2006). MEGAN uses satellite-driven

land cover and modelled meteorological information (e.g., temperature, and photosynthetically available radiation, PAR) to estimate VOCs, $NO_x$ and CO from vegetation at 1 km spatial resolution. Dust emissions are generated online by incorporating Goddard Global Ozone Chemistry Aerosol Radiation and Transport (GOCART) scheme from terrain data and modelled meteorology (Chin et al., 2002). The GOCART scheme, described in detail elsewhere (Ginoux et al., 2001; Zhao et al., 2010,

2013), utilises the information about 10 m wind speed, threshold wind velocity (minimum value to reach for the dust emission to occur) and potential dust source region factors to calculate the dust emission flux. The total dust emission fluxes are calculated by multiplying with an empirical dimensional constant which is taken from Ginoux et al. (2001). The GOCART scheme then distributes the emitted dust particles into 4 size bins (described earlier).

For our evaluation of WRF-Chem performance, hourly simulations are conducted for 01 September to 30 November 2016. September falls within the south-west (SW) monsoon season (its withdrawal typically begins in mid-September) whilst October and November are in the post-monsoon season (Annual report 2016, India Meteorological Department). This permits a comparative assessment of meteorology and air quality between the two seasons. Although 2015-2016 was widely recorded as

subject to a pronounced El Niño event, its effects over India lasted only until the summer of 2016 (India Meteorological Department, Govt. of India Ministry of Earth Sciences, 2017) and therefore should not significantly impact the study period. In terms of general climatology, the 2016 SW monsoon rainfall was recorded to be normal over the country, aside from a deficit in rainfall over parts of northwest India.



## 2.2 Meteorological data

WRF-Chem simulated meteorology is compared with observational networks measuring daily surface weather (Iowa Environmental Mesonet- Automated Surface Observing System; IEM-ASOS Network) and atmospheric soundings (radiosonde observations (RAOB), University of Wyoming). Figure 1b shows the locations of the observation sites from these networks. The data links and access details are given in Table S2.

The IEM-ASOS network is an archive of global automated airport weather observations from weather stations operated by national agencies and airport authorities. Hourly air temperature (T2), relative humidity (RH), wind speed (WS) and wind direction (WD) data for 49 observation sites (Table S4) within the study domain are used. Processing and general quality control checking of the data is undertaken by the IEM-Network so the downloaded data was only checked for missing values before 205   comparison with model output.

Radiosonde measurements for vertical meteorology profile comparison are available for eight sites within the model domain. Pilot balloon soundings are undertaken by the India Meteorological Department and rigorous quality checks are performed before making them freely available (Durre et al., 2006). The radiosonde measurements are available each day at 00:00 UTC (05.30 and 17.30 Indian 210   Standard Time (IST), respectively). No station has complete soundings for the entire study period so model-measurement comparisons include only times when observations are available. The sounding observations are vertically interpolated to the model's pressure levels from 1000 hPa to 100 hPa. The average vertical temperature, virtual potential temperature (VPT), WS and RH profiles are compared for individual sites and temporal variability (as standard deviation) is reported for the entire period 215   across all the pressure levels.

The spatial features of modelled meteorology are compared against the global MERRA-2 reanalysis (Gelaro et al., 2017) dataset available at a latitude-longitude grid resolution of $0.5° \times 0.625°$ and 72-eta hybrid levels at 6-h frequency. MERRA-2 reanalysis data is provided by NASA's Global Modelling and Assimilation Office (GMAO). The meteorological variables are re-gridded to WRF-Chem spatial 220   resolution (12 km) and comparison was undertaken for T2, 10 m WS, water vapour mixing ratio (QV) and planetary boundary layer height (PBLH) variables.

## 2.3 Ground-based PM$_{2.5}$

We evaluate the performance of WRF-Chem in simulating aerosols by comparing modelled PM$_{2.5}$ mass concentrations and aerosol optical depth (AOD) at 550 nm with observations and reanalysis products. 225   The measurements of surface PM$_{2.5}$ used for model comparison are undertaken by the Central Pollution Control Board of India (CPCB), accessed via the OpenAQ platform (Table S1, Fig. 1b). In addition to general quality control procedures applied by CPCB, the hourly PM$_{2.5}$ mass concentration data for 20



stations in the study domain was filtered for missing, zero and negative values. Days with <40 % of hourly measurements were also removed before comparing with the modelled $PM_{2.5}$ mass concentrations. Since Delhi has many more individual sites than other states in the domain, the data is grouped into two categories: all sites within the Delhi region ($n = 8$), and the remaining sites (referred to as 'Others', $n = 12$), the majority of which are located within the IGP region (Fig. 1b, Table S4).

**2.4 Reanalysis $PM_{2.5}$ and Black Carbon concentrations**

The spatial distributions of modelled surface $PM_{2.5}$ and BC concentrations are compared with MERRA-2 global reanalysis products, which is based on the GOCART scheme employed in the Goddard Earth Observing System version 5 (GEOS-5) atmospheric model (Randles et al., 2017). The GOCART model in MERRA-2 employs the online coupling of radiatively-active aerosols with meteorology in the GEOS-5 model. GOCART in MERRA-2 simulates OC, BC, sea salt, dust, and sulfate aerosols which are used to derive the total $PM_{2.5}$ mass concentrations. The aerosols in the GOCART scheme are externally mixed and exclude the treatment of nitrate aerosols (Randles et al., 2017). AOD in MERRA-2 is assimilated using multiple satellite and ground-based observation data including bias-corrected AOD from the Moderate Resolution Imaging Spectroradiometer (MODIS), Advanced Very High-Resolution Radiometer (AVHRR) instruments, Multi-angle Imaging Spectroradiometer (MISR) and Aerosol Robotic Network (AERONET). The aerosol assimilation uses satellite radiance and albedo from observing sensors and bias-corrected AOD, described in detail in Randles et al (2017). Based on past studies and recommendations, the $PM_{2.5}$ concentration is calculated via the following summation of aerosol components in the size bin $\leq 2.5$ µm diameter.

$$[PM_{2.5}] = [BC] + 1.6 \times [OC] + 1.375 \times [SO_4^{2-}] + [Dust] + [Sea\ Salt]$$

The multiplication factor of 1.375 on the sulfate ion concentration is based on the assumption in MERRA-2 that sulfate is primarily present as neutralised ammonium sulfate (Buchard et al., 2016; Provenc̦al et al., 2017; Song et al., 2018). OC in MERRA-2 is scaled up to organic matter concentration using values ranging from 1.2 - 2.6 and this study uses the factor 1.6 which is commonly used for urban carbonaceous particles (Chow et al., 2015; Buchard et al., 2016; Provenc̦al et al., 2017; Song et al., 2018).

**2.5 Satellite and ground-based AOD data**

WRF-Chem AOD at 550 nm is compared with satellite observations from the MODIS sensor on board the Terra and Aqua polar orbiting satellites. The AOD products from MODIS have a 10 km horizontal resolution at equatorial local overpass times of 10.30 (Terra) and 13.30 (Aqua). AOD retrievals from MODIS are based on combined Dark Target (DT: retrieval algorithm over dark land and ocean surfaces) and Dark Blue algorithms (DB: bright land surface) and re-gridded to the WRF-Chem resolution of 12





km. AOD in WRF-Chem is simulated between wavelengths 300 - 1000 nm and interpolated to 550 nm using the Ångström power law (Ångström, 1964; Kumar et al., 2014).

In addition, ground-based AERONET version 2 level 2.0 (quality-assured and cloud-screened) AOD is available at four locations (Fig. 1b) within the study domain and is used for comparison with modelled results. AERONET is a global network (Holben et al., 1998) that has been extensively used for validating satellite observations over South Asia (Sayer et al., 2014; Mhawish et al., 2017).

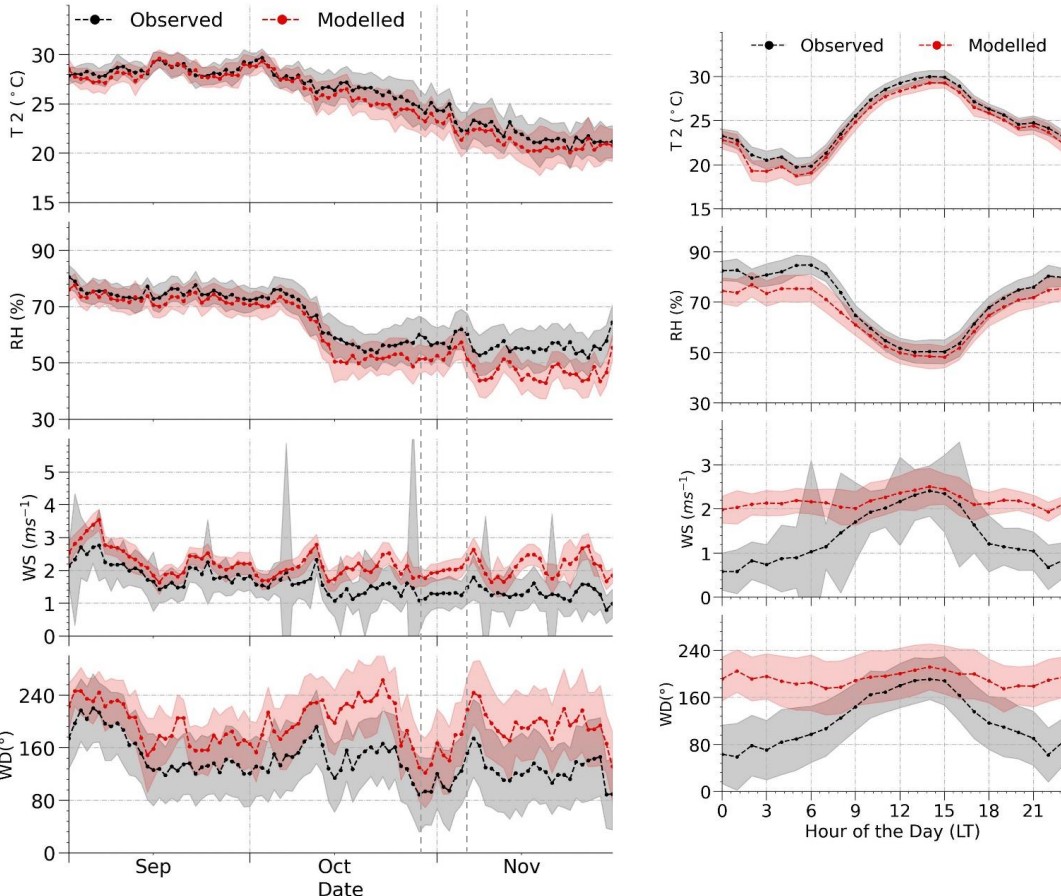

**Figure 2.** Daily-mean time series (left) and mean diurnal cycle (right) of observed (black) and modelled (red) meteorological variables from 01 September – 30 Nov 2016 averaged across the 49 ASOS measurement sites shown in Figure 1b. From top to bottom: daily mean 2-m temperature, relative humidity, wind speed, and wind direction. The shaded regions indicate the standard deviation in the spatial variability in the model and measured variables. The vertical dashed lines delineate the period of severe high pollution





## 2.5 Statistical Metrics

Statistical metrics used here for the evaluation of model performance include mean bias (MB), normalized mean bias (NMB), mean absolute error (MAE), root mean square error (RMSE) and
270 Pearson's correlation coefficient ($r$). Definitions of these metrics are provided in Supplementary Material Table S3.

**Table 1.** Summary of statistical comparison of monthly averaged modelled and observed meteorology variables from September to November 2016 for the 49 ASOS measurement sites shown in Figure 1b. The statistical metrics used for comparison are mean bias (MB), normalized

| Month/Variable | MB | NMB | MAE | RMSE | *r* |
|---|---|---|---|---|---|
| | | (*N* = 49) | | | |
| Temperature (ºC) | | | | | |
| September | -0.28 | -0.01 | 1.5 | 2.2 | 0.86 |
| October | -0.75 | -0.03 | 1.8 | 2.6 | 0.90 |
| November | -0.84 | -0.04 | 2.2 | 3.0 | 0.87 |
| RH (%) | | | | | |
| September | -1.90 | -0.03 | 7.8 | 10.0 | 0.75 |
| October | -4.10 | -0.07 | 10.1 | 13.2 | 0.79 |
| November | -8.20 | -0.15 | 12.8 | 17.7 | 0.65 |
| Wind Speed (m s$^{-1}$) | | | | | |
| September | 0.40 | 0.20 | 0.8 | 1.07 | 0.62 |
| October | 0.54 | 0.36 | 1.0 | 1.26 | 0.30 |
| November | 0.81 | 0.61 | 1.1 | 1.37 | 0.40 |





**Table 2.** Summary of statistical comparison of monthly averaged WRF-Chem and MERRA-2 derived meteorology variables from September to November 2016. The statistical metrics used for comparison are mean bias (MB), normalized mean bias (NMB), mean absolute error (MAE), root mean square error (RMSE) and Pearson's correlation coefficient (*r*).

| Month | MB | NMB | MAE | RMSE | *r* |
|---|---|---|---|---|---|
| Temperature (°C) | | | | | |
| September | -0.57 | -0.03 | 1.4 | 2.12 | 0.99 |
| October | -1.5 | -0.10 | 1.9 | 2.67 | 0.99 |
| November | -2.4 | -0.22 | 2.7 | 3.30 | 0.99 |
| Wind Speed (m s$^{-1}$) | | | | | |
| September | -0.17 | -0.09 | 0.57 | 0.73 | 0.85 |
| October | -0.23 | -0.12 | 0.64 | 0.85 | 0.76 |
| November | -0.17 | -0.08 | 0.79 | 1.09 | 0.73 |
| QV2 (g kg$^{-1}$) | | | | | |
| September | 0.56 | 0.05 | 0.94 | 1.38 | 0.98 |
| October | 0.19 | 0.02 | 0.79 | 1.11 | 0.98 |
| November | -0.07 | -0.01 | 0.65 | 0.99 | 0.97 |
| PBLH (m) | | | | | |
| September | -324 | -0.28 | 355 | 430 | 0.69 |
| October | -477 | -0.37 | 481 | 550 | 0.67 |
| November | -344 | -0.36 | 356 | 446 | 0.70 |
| PM$_{2.5}$ (µg m$^{-3}$) | | | | | |
| September | 54 | 1.9 | 55.1 | 72 | 0.87 |
| October | 20 | 0.49 | 21.7 | 30 | 0.87 |
| November | -8.4 | -0.12 | 13.8 | 23 | 0.95 |
| BC (µg m$^{-3}$) | | | | | |
| September | 0.52 | 0.65 | 0.57 | 0.93 | 0.91 |
| October | 0.24 | 0.19 | 0.44 | 0.79 | 0.91 |
| November | -0.78 | -0.28 | 0.89 | 1.42 | 0.92 |





### 3. Meteorology evaluation results

### 3.1 Near-surface meteorological fields

Figure 2 shows modelled and measured time series of daily means and mean diurnal cycles (right panel) for 2-m temperature (T2), relative humidity (RH), wind speed (WS) and wind direction (WD) derived from hourly data and averaged across all the observational sites. The statistical comparison metrics for the three months are provided in Table 1. As the exact measurement heights at individual sites are not known, the comparisons are made assuming the standard above-ground heights of 2 m for temperature and RH and 10 m for wind speed and direction. Daily average T2 variability correlates well between the model and observations for all the months ($r > 0.85$), with maxima and minima captured well (Fig. 2). Model MB for T2 is very slightly low, but by less than -0.8 ºC for all months. The T2 diurnal profile is also well represented by the model, with differences slightly larger (up to 2 ºC) during night-time.

The general day-to-day variability in modelled surface RH also compares reasonably well with the observations ($r$ range across the months: 0.65 – 0.79) with slight underestimations that gradually increase from -1.9 % in September to -8.2 % in November mainly due to underestimations seen in the night-time RH peaks. The observed diurnal RH cycle is also well simulated by the model, although as for T2 with larger differences during the night when RH is greatest.

The differences in simulated 10 m wind patterns are relatively higher than those for T2 and RH with modelled WS showing a relatively poor correlation of $r \leq 0.4$ and overestimations of about 0.5 – 0.8 m s$^{-1}$ (36 - 61 %) in October and November. However, better correlation ($r = 0.62$) and lower biases (MB = 0.4 m s$^{-1}$ and NMB = 0.2) are observed for September. The diurnal variation of WS during daytime is captured quite well by the model, while the bias high at night (up to 1.5 m s$^{-1}$) is the reason for the observed large biases in modelled daily variabilities in WS. Since local WD is highly variable across sites in different regions it is hard for a model to capture the daily variabilities. As for the other variables, the differences between modelled and observed WD are smallest during the daytime when the general wind direction is south-westerly and largest at night.

Table 2 provides the statistical evaluation results from the comparison of WRF-Chem and MERRA-2 global reanalysis data for mean T2, 10-m wind speed, water vapour mixing ratio (QV) and planetary boundary layer height (PBLH). The spatial maps of these variables are presented in Figures S1 and S2. Except for PBLH the meteorological variables generally show good spatiotemporal agreement between the model and MERRA-2 with the best agreement for T2 and QV, as reflected in the high spatial correlations ($r \geq 0.97$). However, regional heterogeneities exist between the two datasets which are generally more evident temporally across all the variables. The largest spatial differences are seen for WS and QV which show overall underestimations by WRF-Chem for WS (in contrast to observed overestimations as compared to the measured data) and overestimations for QV (in contrast to observed





underestimations as compared to the measured data) in the wider IGP region. There is a stronger west-east gradient in PBLH in MERRA-2 compared to WRF-Chem which possibly influences the $PM_{2.5}$ concentrations in MERRA-2.

A seasonal dry bias in the WRF model over the Indian region due to possible errors in moisture fluxes has been reported previously (Kumar et al., 2012b; Conibear et al., 2018) and night-time

underestimations in modelled RH similar in magnitude to this study were noted by Gunwani and Mohan (2017). A comparison of modelled results including ERA-5 (used to drive WRF-Chem here) and independent MERRA-2 global reanalysis datasets with hourly ground observations (Fig. S3) shows the highest positive bias in RH in ERA-5 during all the months while WRF-Chem and MERRA-2 tend to underestimate RH across all the months. This suggests the model likely these biases likely propagate

from the global reanalysis data used to drive the meteorology. This may affect the model's ability to capture the diurnal evolution of secondary aerosols by hygroscopic growth, particularly during the night.

The observed positive bias in simulated 10 m WS (also seen in Fig. S1 meteorology comparison with ERA-5 and MERRA-2) is well known and the observed magnitude of the bias is largely consistent with

previous studies (Zhang et al., 2016; Mues et al., 2018; Gunwani and Mohan, 2017; Wang et al., 2021). First, this could be in part due to inaccurate land-surface parameterizations (such as roughness length or surface drag and urban canopy) yielding smaller friction velocities and stronger winds in the model. Second, it could also be due to unknown differences in measured and modelled heights. However, the afternoon simulated WS are close to the observations which suggests there are underlying weaknesses

in nocturnal stable boundary layer decoupling in the model. The associated turbulent fluxes and thermodynamic exchanges occurring in the atmospheric boundary layer are important for model simulated PBL and pollutant dispersal (Shen et al., 2023; Nelli et al., 2020). However, during the extreme pollution episode (30 October to 7 November) both model and observations agree on a reduction in WS (although with varying magnitudes) and a shift in WD. These changes highlight the

role of stagnant meteorology in greatly enhancing the near-surface pollution lasting over a week.



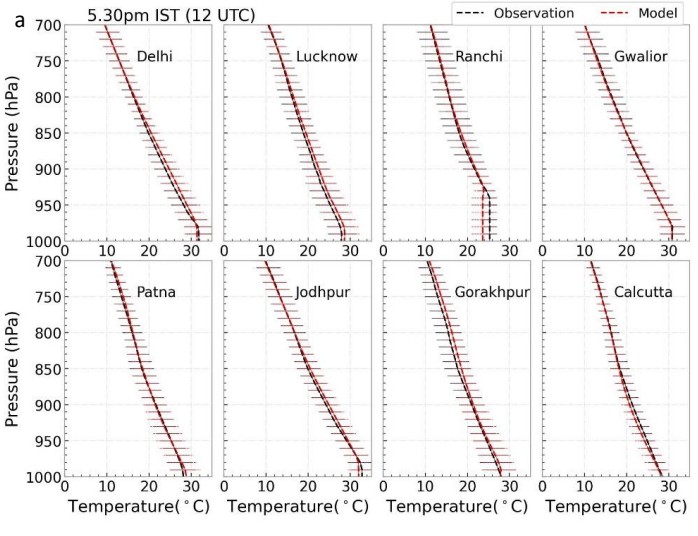

**Figure 4.** Top to bottom**:** Comparisons of vertical profiles of temperature (ᵒC), virtual potential temperature (VPT, ᵒC) and wind speed (m s$^{-1}$) between the model (red) and radiosonde observations (black) for 8 sites at 12 UTC (17.30 IST) averaged for September – November 2016. The horizontal lines show the standard deviation in the day-to-day temporal variability during the comparison period.

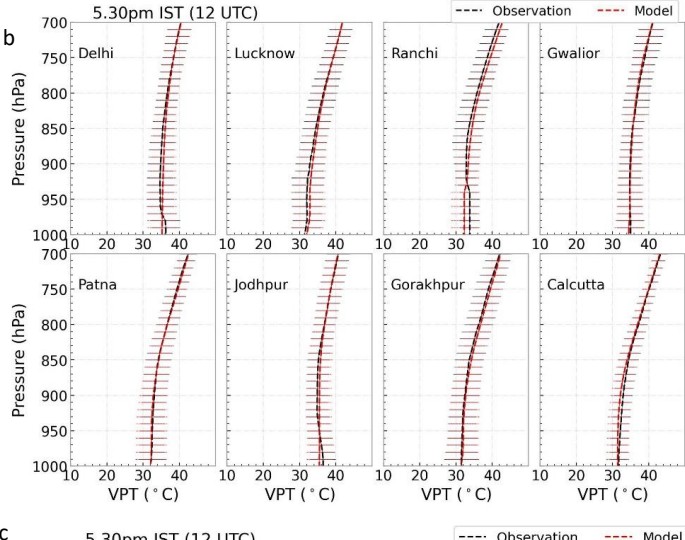

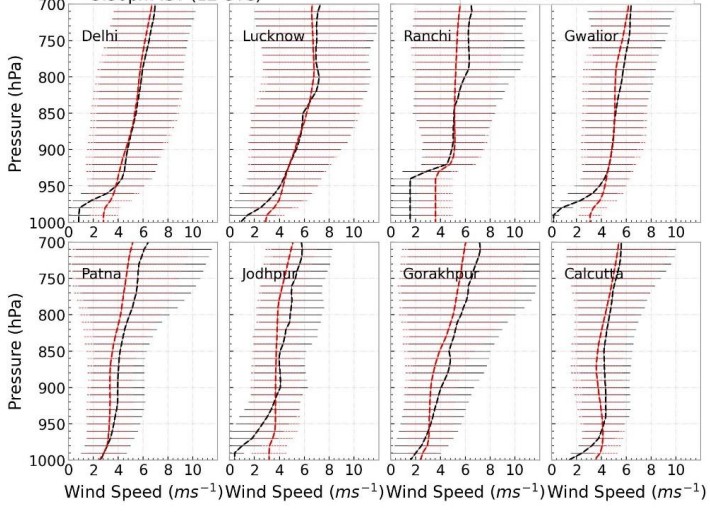



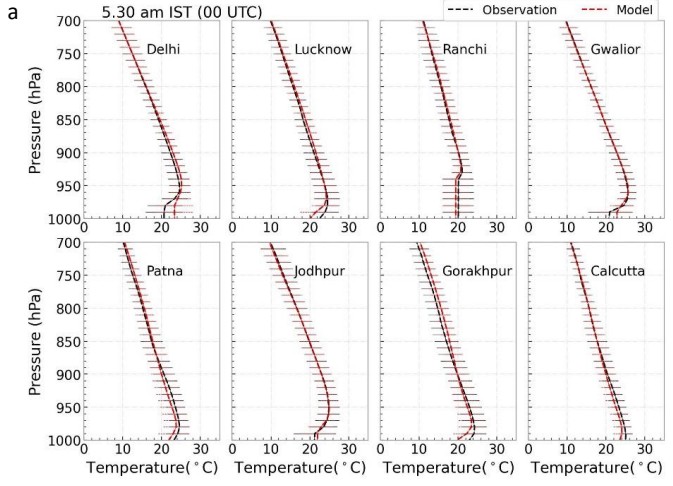

**Figure 3.** Top to bottom: Comparisons of vertical profiles of temperature (°C), virtual potential temperature (VPT, °C) and wind speed (m s$^{-1}$) between the model (red) and radiosonde observations (black) for 8 sites at 00 UTC (5.30 IST) averaged for September – November 2016. The horizontal lines show the standard deviation in the day-to-day temporal

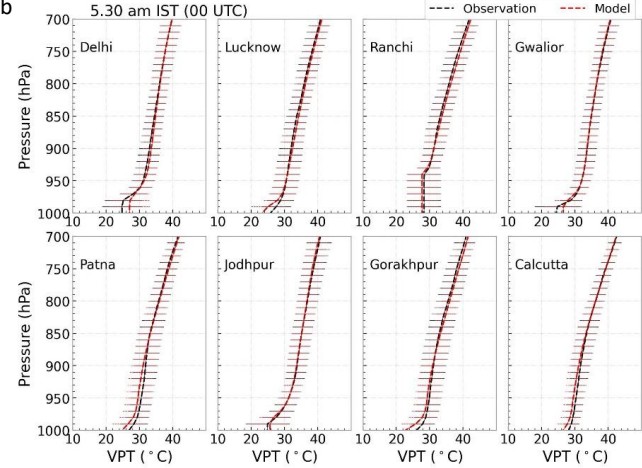

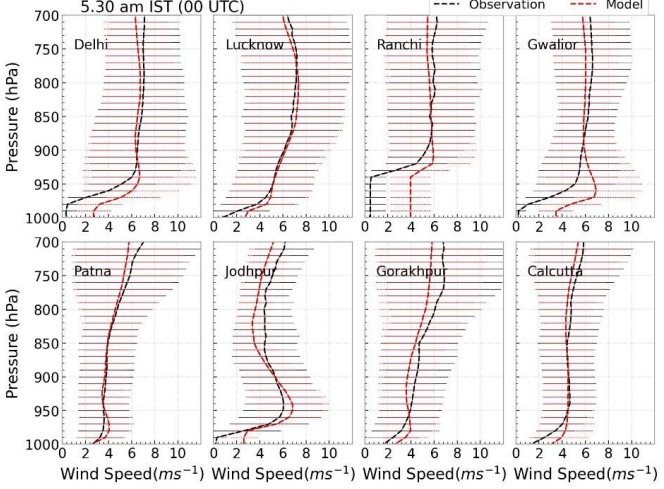





**3. 2 Vertical Profiles**

Figures 3 and 4 show averaged modelled and observed sounding profiles over individual RAOB sites (Fig. 1) for temperature (T), virtual potential temperature (VPT) and wind speed (WS) at 05.30 IST (00 UTC) and 17.30 IST (12 UTC), respectively. The corresponding summaries of statistical metrics are presented in Tables 3 and 4. The upper air meteorology and thermodynamic structure are crucial

parameters of the atmosphere as they impact the transport and convective distribution of pollutants. Of all the meteorological quantities examined here, vertical profiles of T and VPT are represented best by the model, with correlations of $r \geq 0.95$ across all the sites and $r = 1.0$ for most of the sites at both times. At 05.30 IST, modelled T profiles show a warm bias of up to 1.5 ℃ at 6 sites and a cold bias of up to 2 ℃ at Delhi and Gwalior sites up to about 980 hPa (Fig. 3a) which gradually decreases with altitude.

The model also captures the observed marked inversion near the surface in morning T and VPT profiles reasonably well at most sites. Agreement at 17.30 IST is even better (Fig. 4a): biases in modelled T profiles are less than 0.5 ℃ below 980 hPa at all sites except Ranchi and negligible aloft. Overall, across all sites, the average MB, NMB and RMSE values are generally lower for VPT compared to T at both times (Tables 3 and 4).

The simulated WS vertical profiles have larger variations across most of the sites at both times as compared to T and VPT profiles (Figures 3c and 4c). Consistent with the 10 m WS comparisons, the model tends to overestimate WS vertically by up to 4 m s$^{-1}$ at 05.30 IST and up to 3 m s$^{-1}$ at 17.30 IST in the lower layers but better captures it aloft (above ~900 hPa) with only slight differences across all the sites (Fig. S4). Despite the considerable positive bias within the bottom layers, the model reproduces

the observed higher WS at higher altitudes reasonably well, resulting in good correlations of $r \geq 0.77$ at 05.30 IST and $r \geq 0.95$ at 17.30 IST. As an exception, the modelled WS profiles are very well represented over the Patna site (in the east) during both times. The results here differ from those of Mohan and Bhati (2011) who noted increased deviation in simulated WS at higher altitudes over Delhi during the summer months.

The simulated RH profiles were also evaluated (Fig. S4) and show underestimations by up to 20 % in the lower layers of the model across most sites at both times, which decreases in magnitude at higher altitudes except at Gorakhpur. These biases vertically are generally more negative at 05.30 IST compared to 17.30 IST indicating a dry bias in early morning hours in the model, consistent with the ground observation comparisons.

VPT profiles are particularly useful in understanding the stability and turbulence of the atmosphere which helps in the dilution of the pollutants within the mixed boundary layer. By accounting for moisture and temperature, a VPT profile indicates buoyancy and stability in the atmosphere and can be used to derive planetary boundary layer heights (Liu et al., 2019; Vogelezang and Holtslag, 1996). Figure 3 shows that, at all sites, observed and simulated temperature inversion layers close to the surface



**Table 3.** Summary of statistical comparison of modelled and observed 0.5.30 IST profiles from radiosonde data for the individual RAOB stations shown in Figure 1b averaged from September to November 2016. The statistical metrics used for comparison are mean bias (MB), normalized mean bias (NMB), mean absolute error (MAE), root mean square error (RMSE) and Pearson's correlation coefficient ($r$).

| Station Name | MB | NMB | MAE | RMSE | $r$ |
|---|---|---|---|---|---|
| **Temperature (°C)** | | | | | |
| Calcutta | -0.22 | 0.03 | 0.58 | 0.83 | 1.00 |
| Delhi | 0.14 | -0.02 | 0.67 | 1.01 | 1.00 |
| Gorakhpur | 0.14 | -0.02 | 1.14 | 1.68 | 1.00 |
| Gwalior | -0.08 | 0.01 | 0.67 | 1.12 | 1.00 |
| Jodhpur | -0.25 | 0.03 | 0.92 | 1.92 | 1.00 |
| Lucknow | -0.81 | 0.10 | 1.54 | 7.70 | 0.96 |
| Patna | -0.15 | 0.02 | 0.75 | 1.06 | 1.00 |
| Ranchi | -1.08 | 0.14 | 1.70 | 7.56 | 0.96 |
| **VPT (°C)** | | | | | |
| Calcutta | -0.31 | -0.01 | 0.70 | 0.98 | 1.00 |
| Delhi | 0.09 | 0.00 | 0.74 | 1.09 | 1.00 |
| Gorakhpur | -0.02 | 0.00 | 1.37 | 1.99 | 0.99 |
| Gwalior | -0.16 | 0.00 | 0.78 | 1.29 | 1.00 |
| Jodhpur | -0.37 | -0.01 | 1.13 | 2.65 | 0.99 |
| Lucknow | 0.14 | 0.00 | 1.24 | 3.87 | 0.98 |
| Patna | -0.27 | -0.01 | 0.90 | 1.25 | 1.00 |
| Ranchi | 0.68 | 0.01 | 1.70 | 5.99 | 0.95 |
| **WS (m s$^{-1}$)** | | | | | |
| Calcutta | -0.35 | -0.04 | 1.34 | 2.02 | 0.96 |
| Delhi | -0.29 | -0.02 | 1.57 | 2.08 | 0.99 |
| Gorakhpur | -0.78 | -0.07 | 1.80 | 2.42 | 0.98 |
| Gwalior | -0.33 | -0.03 | 1.60 | 2.37 | 0.97 |
| Jodhpur | -0.74 | -0.06 | 1.82 | 2.41 | 0.98 |
| Lucknow | 0.15 | 0.01 | 2.37 | 4.92 | 0.89 |
| Patna | -0.59 | -0.06 | 1.53 | 2.13 | 0.98 |
| Ranchi | 0.58 | 0.07 | 2.65 | 5.33 | 0.77 |
| **RH (%)** | | | | | |
| Calcutta | -1.09 | -0.02 | 7.56 | 11.9 | 0.93 |
| Delhi | -2.20 | -0.08 | 6.23 | 10.0 | 0.92 |
| Gorakhpur | -10.5 | -0.21 | 15.4 | 19.4 | 0.88 |
| Gwalior | -1.87 | -0.06 | 7.21 | 10.9 | 0.93 |
| Jodhpur | 1.54 | 0.07 | 8.59 | 11.7 | 0.89 |
| Lucknow | -2.99 | -0.08 | 10.4 | 15.1 | 0.87 |
| Patna | -0.84 | -0.02 | 9.37 | 14.4 | 0.92 |
| Ranchi | -3.12 | -0.07 | 10.6 | 16.5 | 0.89 |





at 5.30 IST, demonstrating the typical formation of an urban nocturnal stable boundary layer. In contrast, at 17.30 IST (Fig. 4), both the observed and modelled VPTs exhibit a typical well-mixed late afternoon profile due to surface heating, with higher values of VPT near the surface (33–36 ºC surface) that remains nearly constant up to about 850 hPa across most sites. The negligible biases and error statistics in T and VPT profiles (Tables 3 - 4) across all sites provide high confidence in model skill in simulating the thermodynamic structure of the atmosphere. This is an improvement on Mues et al. (2018) who reported larger biases in T profiles (up to 3 ºC and 7 ºC at 05.30 IST and 17.30 IST, respectively) at the Delhi site in winter and summer 2013. As noted in Section 3.1, and elsewhere (Mohan and Bhati, 2011; Gunwani and Mohan, 2017), errors in simulated WS are highly sensitive to local roughness length and model topography and are thus subject to greater noise. Given these limitations, we find the model performance statistics comparable to previous studies (Mohan and Bhati, 2011; Kumar et al., 2012b) and close to the benchmarks provided by Emery and Tai (2001).

## 4 Chemistry evaluation results

### 4.1 Ground-based PM$_{2.5}$

Figure 5 compares the modelled and measured daily averaged time series (left) and diurnal variability (right) of surface PM$_{2.5}$ concentrations from hourly samples from September to November 2016. The observations are spatially averaged across 8 sites in Delhi and 12 sites across the rest of the domain (referred to as 'Others'). The statistical summary is presented in Table 5. The model adequately captures the day-to-day variation of PM$_{2.5}$ for October-November, when it is biased low, while it fails to reproduce the daily variability during September when it is strongly biased high. On average, during September the model overestimates surface PM$_{2.5}$ concentrations by more than a factor of two (NMB range: 1.69 to 1.91) across all the sites and underestimates in November by 26 % over Delhi and by 14 % over Others. Overall, the model and observed daily surface PM$_{2.5}$ correlate reasonably well during October (Delhi: $r = 0.65$, Others: $r = 0.53$) and November (Delhi: $r = 0.76$, Others: $r = 0.66$). Correlation for these months is better across Delhi sites but shows relatively larger mean biases (+17.7 to – 73.2 µg m−3) and NMBs (+0.13 to – 0.26) compared to Others. Additionally, the model tends to predict PM$_{2.5}$ concentrations with a fairly broad range of monthly RMSE values (56.3 – 138 µg m$^{-3}$).

The spatially averaged diurnal cycle for modelled surface PM$_{2.5}$ shows a pronounced diurnal trend matching observations for Delhi sites, while the diurnal cycle is less pronounced at Others sites. Generally, diurnal trends are in good agreement across all sites, although on average the model tends to underpredict the afternoon dips and night-time peaks compared to the observations, indicating missing anthropogenic activities from the simplified emissions patterns derived from monthly estimates used in the model. The lack of a representation of a realistic diurnal activity cycle in the anthropogenic emissions highlights meteorology could be driving the modelled PM$_{2.5}$ variation. Although this might partly be affected by the imperfectly represented diurnal variability of WS in the model (Section 3.1).



**Table 4.** Same as Table 3, but for 17.30 IST profiles.

| Station Name | MB | NMB | MAE | RMSE | r |
|---|---|---|---|---|---|
| **Temperature (ºC)** | | | | | |
| Calcutta | -0.16 | 0.02 | 0.57 | 0.82 | 1.00 |
| Delhi | 0.04 | -0.01 | 0.63 | 0.85 | 1.00 |
| Gorakhpur | -0.02 | 0.00 | 1.24 | 1.89 | 1.00 |
| Gwalior | -0.17 | 0.02 | 0.68 | 1.15 | 1.00 |
| Jodhpur | -0.11 | 0.01 | 0.74 | 1.02 | 1.00 |
| Lucknow | 0.01 | 0.00 | 0.95 | 1.73 | 1.00 |
| Patna | 0.06 | -0.01 | 0.75 | 1.20 | 1.00 |
| Ranchi | -1.22 | 0.18 | 2.15 | 8.45 | 0.96 |
| **VPT (ºC)** | | | | | |
| Calcutta | -0.24 | 0.00 | 0.68 | 1.00 | 1.00 |
| Delhi | -0.02 | 0.00 | 0.74 | 0.97 | 1.00 |
| Gorakhpur | -0.31 | -0.01 | 1.50 | 2.19 | 0.99 |
| Gwalior | -0.26 | 0.00 | 0.81 | 1.39 | 1.00 |
| Jodhpur | -0.20 | 0.00 | 0.91 | 1.25 | 1.00 |
| Lucknow | -0.12 | 0.00 | 1.16 | 2.13 | 0.99 |
| Patna | -0.01 | 0.00 | 0.85 | 1.34 | 1.00 |
| Ranchi | 0.65 | 0.01 | 1.89 | 5.59 | 0.96 |
| **WS (m s$^{-1}$)** | | | | | |
| Calcutta | -0.30 | -0.04 | 1.30 | 1.78 | 0.97 |
| Delhi | -0.24 | -0.02 | 1.48 | 1.91 | 0.99 |
| Gorakhpur | -0.71 | -0.06 | 1.76 | 2.32 | 0.98 |
| Gwalior | -0.27 | -0.02 | 1.50 | 1.97 | 0.98 |
| Jodhpur | -0.64 | -0.06 | 1.76 | 2.27 | 0.98 |
| Lucknow | -0.40 | -0.03 | 1.81 | 2.42 | 0.98 |
| Patna | -0.68 | -0.06 | 1.49 | 2.01 | 0.98 |
| Ranchi | -0.15 | -0.02 | 1.88 | 2.74 | 0.95 |
| **RH (%)** | | | | | |
| Calcutta | -2.13 | -0.04 | 8.06 | 12.7 | 0.93 |
| Delhi | -1.47 | -0.06 | 6.73 | 10.6 | 0.90 |
| Gorakhpur | -9.62 | -0.20 | 14.9 | 18.8 | 0.86 |
| Gwalior | -0.58 | -0.02 | 7.80 | 12.0 | 0.90 |
| Jodhpur | 2.29 | 0.12 | 8.54 | 11.4 | 0.86 |
| Lucknow | -2.12 | -0.06 | 10.9 | 14.9 | 0.85 |
| Patna | -0.96 | -0.02 | 9.44 | 14.0 | 0.91 |
| Ranchi | 1.73 | 0.04 | 10.3 | 15.3 | 0.89 |



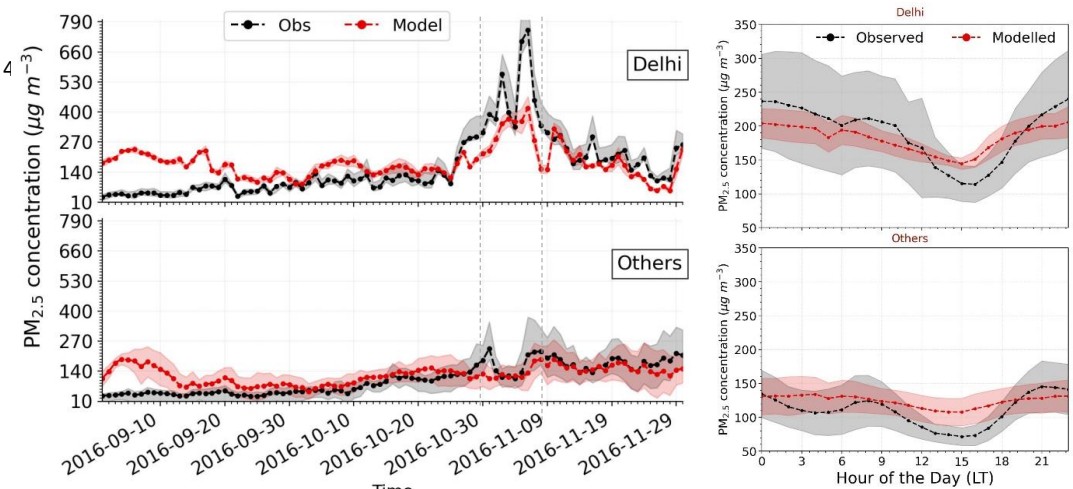

**Figure 5.** Time series of daily means (left) and mean diurnal cycles (right) of observed and modelled PM$_{2.5}$, averaged across 8 sites in Delhi and 12 sites over the rest of the domain (labelled 'Others') from September – November 2016. The shaded area in both panels shows standard deviation of the spatial variability of the model and measured PM$_{2.5}$. The locations of the ground measurement sites are shown in Figure 1b. The vertical dashed lines delineate the period of severe high pollution between 30 October and 7 November.

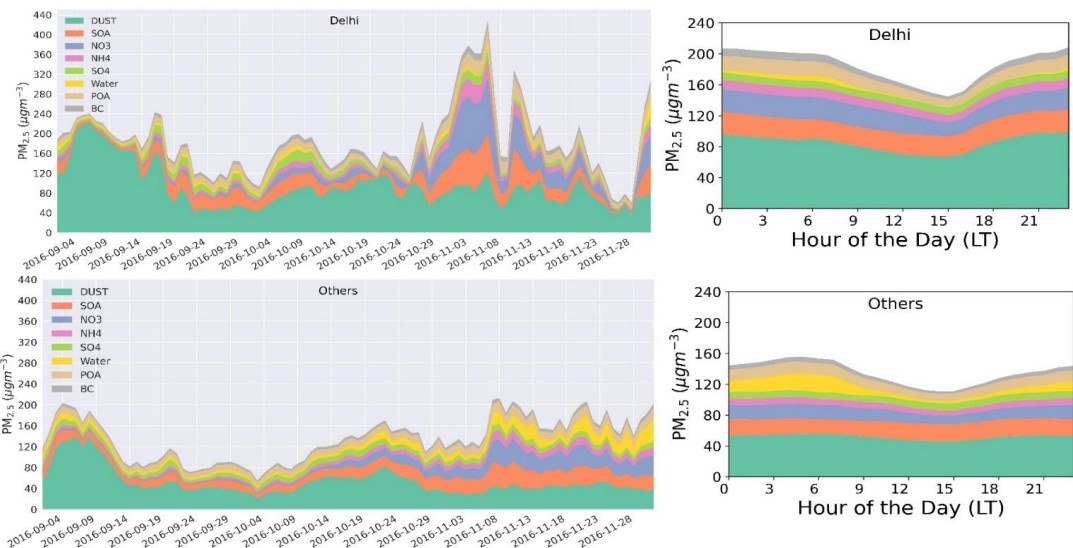

**Figure 6.** Time series of daily means (left) and mean diurnal cycles (right) of modelled individual PM$_{2.5}$ components averaged across 8 stations in Delhi and 12 stations over the rest of the domain (labelled 'Others') from September – November 2016. The individual species contribution abbreviations are: SOA (secondary organic aerosol), POA (primary organic aerosol), SO$_4^{2-}$ (sulfate), NH$_4^+$ (ammonium), NO$_3^-$ (nitrate), BC (black carbon). The vertical dashed lines delineate the period of severe high pollution between 30 October and 7 November.



During the 30 October –7 November pollution episode both observations and model show the highest daily mean surface $PM_{2.5}$ (Observed: $300 - 750$ µg m$^{-3}$, Modelled: $150 - 420$ µg m$^{-3}$) across Delhi, while relatively lower concentrations are seen across Others sites during this period (observed and modelled: $< 200$ µg m$^{-3}$) (Fig. 5). The observed daily mean $PM_{2.5}$ concentrations exceed the 24-h
average 2021 WHO air quality guideline of 15 µg m$^{-3}$ (WHO 2021) by nearly 50 times and the predicted concentrations exceed by nearly 28 times. The maximum negative differences (up to 350 µg m$^{-3}$) between the daily mean modelled and observed $PM_{2.5}$ also occur during this episode. During this period, the observed hourly $PM_{2.5}$ concentrations exceed 1000 µg m$^{-3}$ (mostly at night) at the Delhi US embassy site (in central Delhi) and exceed 800 µg m$^{-3}$ at all the sites across Delhi and two downwind stations in
the lower IGP (Lucknow and Kanpur). The corresponding modelled hourly concentrations at these locations and times underestimate $PM_{2.5}$ by a factor of 2-3 ($380 - 520$ µg m$^{-3}$), in part attributable to overestimated surface WS. One study characterising this 2016 high pollution episode over Delhi reported exceptionally high night-time mean $PM_{2.5}$ concentrations of 2924 µg m$^{-3}$ on 30 October (Diwali festival night), 1520 µg m$^{-3}$ on 5 November, and day time mean values of nearly 1500 µg m$^{-3}$
on 6 November (Sawlani et al., 2019). The modelled and observed daily average $PM_{2.5}$ across downwind Others sites peaks ($> 250$ µg m$^{-3}$) only towards the end of the high pollution episode, suggesting a regional distribution of $PM_{2.5}$ over time. The observed and simulated near-surface meteorology during this time over northern India shows stagnant conditions conducive for the build-up of pollutants: smaller WS (1-1.5 m s$^{-1}$), lower PBLH ($< 500$ m) and a nearly 2 - 3 ºC drop in near-surface temperature leading
to atmospheric inversion (Fig. 2). These stagnant conditions combined with regional and local anthropogenic emissions facilitate pollution accumulation within the shallow continental boundary layer over wider northern India. After the extreme pollution days (9 November onwards), the model captures the magnitude of daily $PM_{2.5}$ variation well everywhere except for an observed peak across Delhi on 17 November.

**4.2 Modelled $PM_{2.5}$ Composition**

The daily time series and average diurnal variability of modelled mean surface $PM_{2.5}$ composition over observation sites in Delhi and Others are shown in Figure 6. Due to the lack of observed $PM_{2.5}$ speciation data for this period, only modelled results are presented here. These are qualitatively compared with literature for other years as the aerosol loading over the Indian region exhibits stronger intra-annual
variabilities than interannual variabilities (Conibear et al., 2018; Mhawish et al., 2021). The largest variations in daily $PM_{2.5}$ components across all months are observed for secondary organic aerosol (SOA) and secondary inorganic aerosol (SIA) (sulfate, nitrate and ammonium) over all the sites. The concentration of fine dust particles dominates most evidently at the beginning of September and reduces to almost half in October and November but remains a non-negligible contributor to total $PM_{2.5}$ on
average (15 - 25 %) across all sites. The fine dust component is mainly responsible for the overestimations seen in modelled $PM_{2.5}$ in September compared to the measurement. Another notable





**Table 5.** Statistical summary of comparisons of monthly mean modelled and observed PM$_{2.5}$ concentrations for September to November 2016 for Delhi (top) and Other stations (bottom). The statistical metrics are mean bias (MB), normalized mean bias (NMB), mean absolute error (MAE), root mean square error (RMSE) and Pearson's correlation coefficient (*r*). *n* denotes

| Month | MB | NMB | MAE | RMSE | r | Obs_mean | Mod_mean |
|---|---|---|---|---|---|---|---|
| **PM$_{2.5}$ (µg m$^{-3}$)** | | | **Delhi sites (*n* = 8)** | | | | |
| September | 111 | 1.91 | 111 | 124 | 0.17 | 58.7 | 170 |
| October | 17.7 | 0.13 | 58.1 | 74.2 | 0.65 | 141 | 159 |
| November | -73.2 | -0.26 | 95.2 | 138 | 0.76 | 279 | 206 |
| | | | **Others sites (*n* = 12)** | | | | |
| September | 69.9 | 1.69 | 70.26 | 89.5 | 0.44 | 41.3 | 111 |
| October | 10.9 | 0.11 | 40.71 | 56.3 | 0.53 | 102 | 113 |
| November | -23.8 | -0.14 | 54.94 | 73 | 0.66 | 172 | 148 |

**Table 6.** Statistical summary of comparisons of monthly mean concentrations (µg m$^{-3}$) of PM$_{2.5}$ and black carbon from the WRF-Chem model and MERRA-2 from September to November 2016. The statistical metrics are mean bias (MB), normalized mean bias (NMB), mean absolute error (MAE), root mean square error (RMSE) and Pearson's correlation coefficient (*r*).

| Month | MB | NMB | MAE | RMSE | r |
|---|---|---|---|---|---|
| | | **PM$_{2.5}$ (µg m$^{-3}$)** | | | |
| September | 54 | 1.9 | 55.1 | 72 | 0.87 |
| October | 20 | 0.49 | 21.7 | 30 | 0.87 |
| November | -8.4 | -0.12 | 13.8 | 23 | 0.95 |
| | | **BC (µg m$^{-3}$)** | | | |
| September | 0.52 | 0.65 | 0.57 | 0.93 | 0.91 |
| October | 0.24 | 0.19 | 0.44 | 0.79 | 0.91 |
| November | -0.78 | -0.28 | 0.89 | 1.42 | 0.92 |



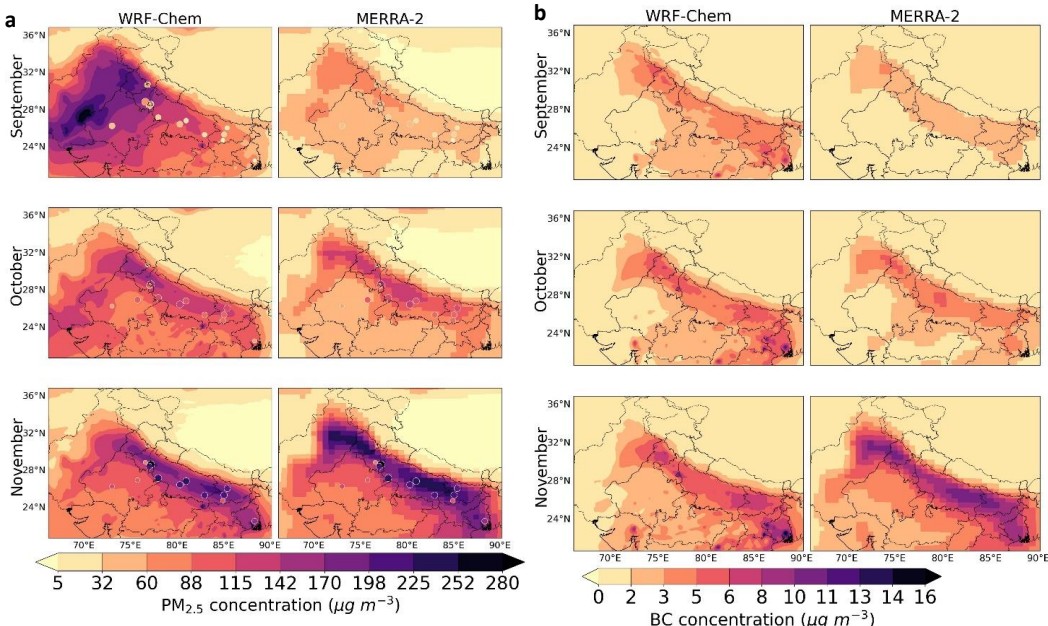

**Figure 7.** Spatial distributions of monthly mean concentrations (µg m$^{-3}$) of a) PM$_{2.5}$ and b) black carbon from the WRF-Chem model and MERRA-2 for September to November 2016. The monthly mean PM$_{2.5}$ at the measurement sites are shown in circles in a).



change is in the nitrate component which dramatically peaks during the high pollution period, together with SOA, ammonium and primary aerosols (OC, BC). The modelled peaks in $PM_{2.5}$ and its components largely follow the observed $PM_{2.5}$ trend (Oct - Nov period) which highlights the model's skill in representing the diversity of aerosols during dramatic shifts in surface particle pollution and is more clearly seen across Delhi sites than Others. Among SIA, the $PM_{2.5}$ composition in November is dominated by nitrate aerosols (10 - 30 %) which are comparable to reported measurements. For example, a high nitrate fraction (20 - 27 %) in post-monsoon months has been reported in various measurement studies over India (Ram and Sarin, 2011; Schnell et al., 2018; Patel et al., 2021; Talukdar et al., 2021). The average modelled BC contribution over Delhi during September (3 µg m$^{-3}$), October (8.2 µg m$^{-3}$) and November (13.2 µg m$^{-3}$) are comparable to the measured EC (assumed to be equivalent to modelled BC) concentrations (~3 µg m$^{-3}$, ~ 6 µg m$^{-3}$ and ~12 µg m$^{-3}$, respectively) reported by Sharma et al. (2018). The dominance of secondary particle contribution to modelled $PM_{2.5}$ during post-monsoon months is fully consistent with other studies (Gani et al., 2019; Talukdar et al., 2021) although the relative abundance is lower. The diurnal variation of $PM_{2.5}$ components over Delhi show more pronounced dips in primary and secondary inorganics, suggesting influence of local emissions while the fine dust component remains relatively stable, suggesting both local and natural non-local emissions influence.

**4.3 Comparison of $PM_{2.5}$ and Black Carbon distribution with reanalysis products**

Figure 7 compares the monthly averaged spatial distribution of WRF-Chem modelled and MERRA-2 reanalysis derived surface $PM_{2.5}$ and BC concentrations. The corresponding domain-averaged performance statistics are summarised in Table 6. The overall spatial agreement between the model and MERRA-2 is excellent for both $PM_{2.5}$ and BC ($r > 0.87$, Fig. S6). However, on a regional scale, the modelled $PM_{2.5}$ is biased high over parts of arid western India and eastern Pakistan in September, resulting in a domain-wide NMB of 1.9. The model shows a stronger west-east gradient in $PM_{2.5}$ than MERRA-2 with the highest modelled concentrations of >250 µg m$^{-3}$ in the western and north-western regions. Agreement between the model and MERRA-2 improves for October-November.

The high simulated $PM_{2.5}$ loading over some parts of north-western India during September is most likely due to erroneous dust uplift by overestimated winds from the Thar Desert in the west (Fig. S6), the major seasonal natural dust source region (Bali et al., 2021; Kumar et al., 2018a). This overestimation could further be enhanced by the underestimation of dust deposition in the model arising from a dry bias over the land region in the domain (Ratnam and Kumar, 2005; Conibear et al., 2018). The notable change in modelled $PM_{2.5}$ over the dust source region along the western borders from September to November shows a strong seasonality in dust emissions in the model. Compared to WRF-Chem, MERRA-2 shows a slightly better comparison with monthly mean surface $PM_{2.5}$ (Fig. 7a) for



individual monitoring sites with smaller differences between model-measured mean than WRF-Chem

(especially for September).

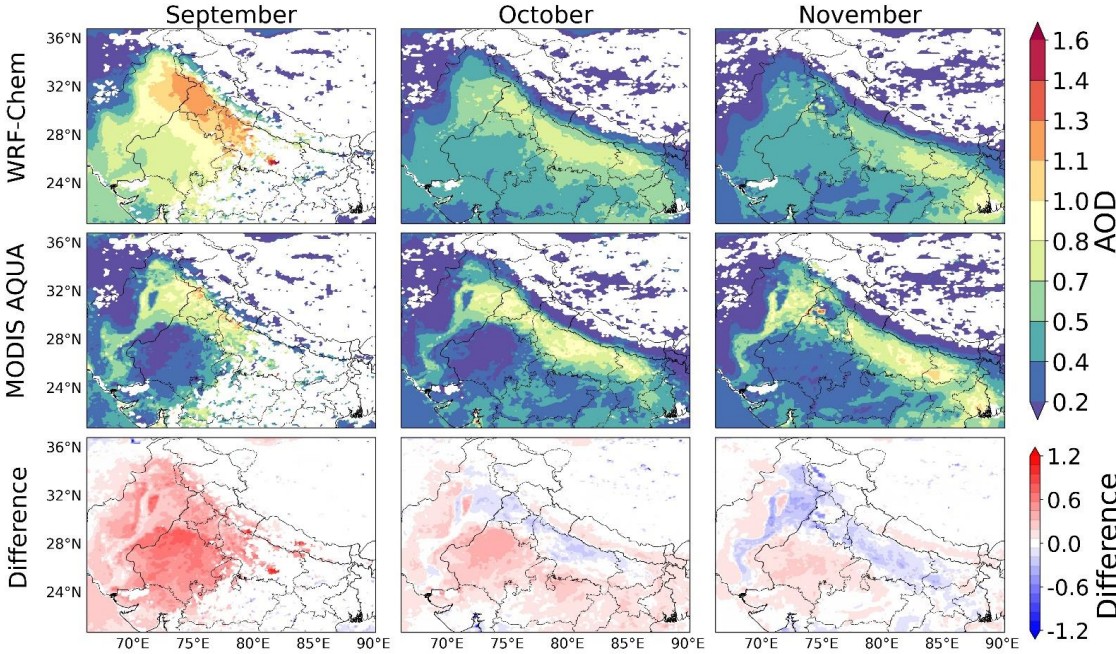

**Figure 8**. Spatial variation of monthly mean AOD at 550 nm derived from the model and MODIS sampled at local overpass times of 10.30 (Terra) and 13.30 (Aqua) for September to November 2016. Absolute differences of model minus satellite AOD are shown in the bottom row.

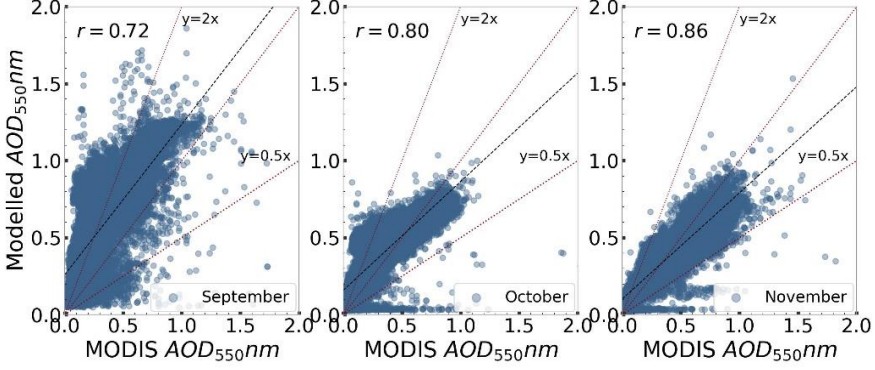

**Figure 9**. Scatterplots of monthly averaged model versus MODIS-derived AOD at 550 nm for the months (from left to right) September, October, and November 2016. The 2:1, 1:1 and 1:2 lines (red dashed lines), the best-fit line (black line) and Pearson's correlation coefficient $r$ are also shown for each month.




In contrast, the highest BC concentrations occur along the IGP for all the months and increases from September to November (Fig. 7b). During October and November, the northwest and eastern parts of the IGP exhibit the highest $PM_{2.5}$ and BC concentrations in both datasets. Compared to MERRA-2, modelled BC shows more distinguishable spatial features including localised hotspots coinciding with densely populated major metropolitan and industrial cities with clusters of coal-fired power plants (Singh et al., 2018). For instance, conspicuous localised regions appear over dense urban centres like Ahmedabad, Delhi, Kolkata, the steel industrial city of Jamshedpur, Raipur with heavy mining, Singrauli with ore-processing industries in the upper central domain, and Jharia coal belts in the east having clusters of coal-fired power plants. Overall, the spatial variabilities of BC and $PM_{2.5}$ are quite similar in both WRF-Chem and MERRA-2 with WRF-Chem estimating slightly lower $PM_{2.5}$ and BC in November over the majority of the IGP except over Delhi.

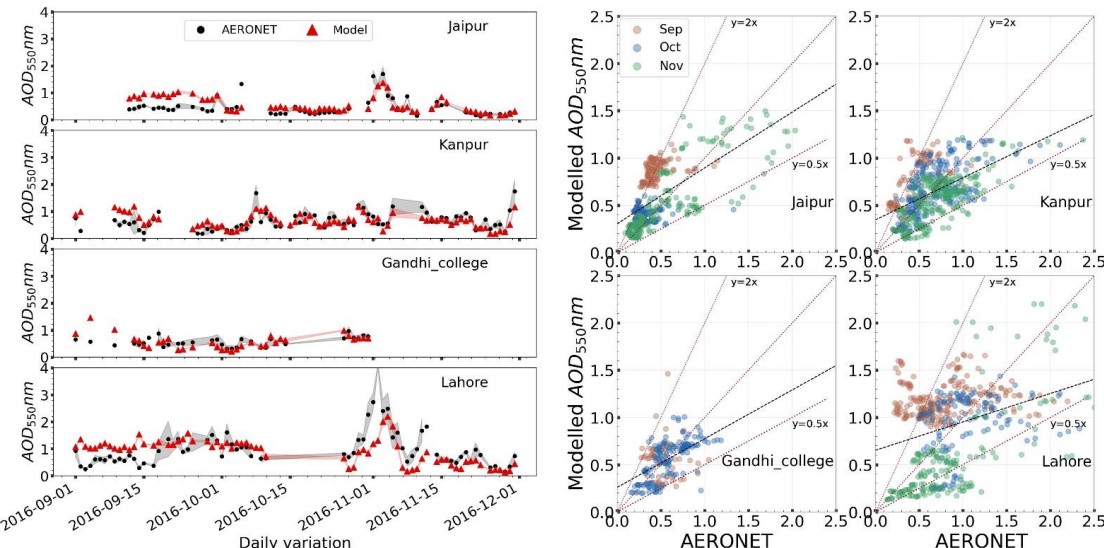

**Figure 10.** Time series (left panel) and scatter plots (right panel) of modelled and AERONET daily averaged AOD at 550 nm over the 4 AERONOET stations shown in Figure 1b for the period September to November 2016.

### 4.4 Evaluation of aerosol optical depth with satellite and AERONET observations

Figure 8 compares WRF-Chem simulated and MODIS (Aqua) retrieved monthly averaged distribution of AOD at 550 nm. The unitless quantity AOD is a measure of particle extinction within the atmospheric column from the surface to the top atmosphere and provides a useful spatial estimate of particle loading using satellite instruments. The spatial distributions of modelled and MODIS AOD agree well for all



months ($r \geq 0.72$ Table 7) although regional biases similar to the MERRA-2 comparisons occur over
northern and western parts of the domain. As with MERRA-2 PM$_{2.5}$ comparisons, during September
the model captures well the high AOD (up to 1.2) over north-western India and along the borders with
north-east Pakistan but predicts higher AOD over the western arid region (Fig. 8), indicated by the
overall NMB of 0.69. The statistical evaluation metrics for all the months (Table 7) show there is a
good overall agreement between modelled and satellite AOD which gradually improves from
September to November. In both model and satellite data, AOD values are generally low (<0.5) outside
of the broader IGP region in all the months. Although the satellite AOD shows higher spatial variability,
a good spatial correlation exists between the two datasets in October-November ($r = 0.80$ and 0.86,
respectively) (Fig. 9). The domain averaged modelled AOD (0.39 and 0.34, respectively) during these
months are comparable to satellite retrieved AOD (0.32 and 0.34, respectively). Despite the overall
underestimations during the biomass burning period of (mid-October to mid-November), the model
captures high AOD values over some small, localised parts in Punjab and Haryana in northern India
and north-eastern Pakistan, although with slightly lower magnitudes. The higher AOD along the entire
IGP region is more apparent from the satellite observations in November, which show AOD values
reaching ~ 2.0 (underestimated in the model by about 10 %) over parts of Punjab in the north and Uttar
Pradesh and Bihar in the east (AOD >1.8). Interestingly, the regional hotspots along the IGP region,
over eastern Uttar Pradesh and eastern Bihar as observed in modelled PM$_{2.5}$ maps during October-
November are evident in MODIS AOD distribution but less discernible in modelled AOD maps (Fig.
8). It is important to note that the MODIS satellite overpass times of 10.30 and 13.30 local time limits
comparisons to the afternoon each day. Therefore, it is the modelled meteorological conditions typical
to daytime (deep PBL height, increased WS) that affect the modelled AOD column. In a similar model
set-up over northern India, Roozitalab et al. (2021) and Kulkarni et al. (2020) found comparable
estimates of modelled AOD distribution during the 2017 post-monsoon high pollution event.

To further evaluate model skill in predicting the optical properties of aerosols, the modelled daily
averaged AOD at 550 nm is compared in Figure 10 against the four AERONET sites (Fig. 1b) in the
study domain. There are missing data at all the sites with Kanpur in the east and Jaipur in the west (both
dense urban locations) having the most data coverage. The daily variabilities of AOD comparison with
point observations show similar trends as previously noted for comparison with satellite AOD and
ground-based and MERRA-2 PM$_{2.5}$ comparisons. The model evaluation against AERONET AOD
largely agrees with the PM$_{2.5}$ evaluations including higher disparities seen for September with a positive
MB (0.02 to 0.43) across all the sites. However, the high daily averaged AERONET AOD (>1.0) at all
sites during the high pollution event at the start of November is captured reasonably well by the model
except in Lahore, a large city in eastern Pakistan, where the model underestimates AOD the most. Of
the four sites, crop residue burning occurs in Lahore (Kulkarni et al., 2020), which is also situated close



to other biomass-burning regions of northwestern India. This AERONET site shows the highest
observed (~3.0) and modelled (~2.0) AOD values during the high pollution episode.

To further check for consistency between satellite and ground measurements, the time series of satellite,
AERONET and modelled AOD at 550 nm at the four observation locations are shown in Figure 11. To
compare the three datasets, the data points corresponding to the local overpass time of MODIS are
selected from the hourly AERONET and WRF-Chem datasets. The satellite AOD generally matches
more closely with AERONET at lower values and misses the magnitude of high AOD during high
pollution days. Earlier studies have attributed the inaccuracies in MODIS AOD retrievals due to dense
haze hanging over north India and the IGP region during severe pollution days (Mhawish et al., 2022).
The modelled AOD captures the hourly AOD trend quite well but also underestimates AOD in absolute
magnitude during high pollution days across the sites. Overall, the modelled AOD agrees well with
satellite and ground observations during October and November despite some underestimations in
absolute magnitudes.

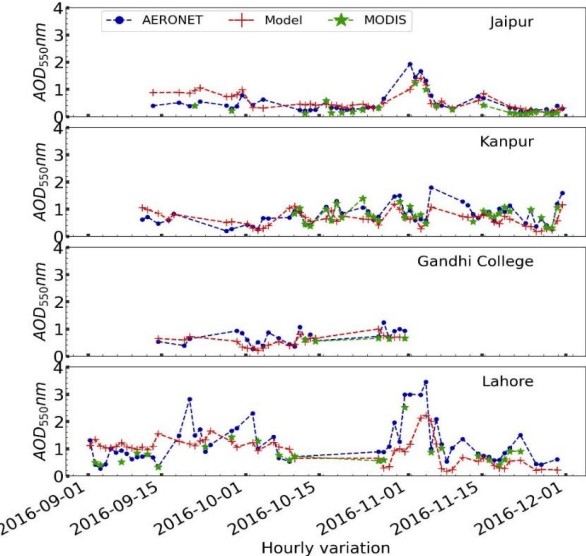

**Figure 11.** Time series of MODIS-retrieved (green), modelled (red) and AERONET (blue) AOD at 550
nm sampled at 13.30 IST over the 4 AERONET stations shown in Fig. 1b: from top to bottom, Jaipur,

**4.5 Discussion**

The discrepancies in model-observation particulate matter comparisons for September have also been
noted in other studies for India and suggest inaccuracies in modelling moisture transport during the




monsoon season, which affects particle deposition and washout (Conibear et al., 2018; Mogno et al., 2021). Furthermore, in 2016, almost all the ground stations were in urban locations of the IGP region, which prevents the evaluation of the model at more spatially representative rural locations. In addition,
nearly all the measurement sites are on or near the roadside with heavy influences from dense traffic and local anthropogenic activities together with dense micro-scale urban features that are challenging to resolve in the model, including because emissions are only at monthly temporal resolution. The sudden jumps in particulate matter during an extreme pollution event are especially difficult to capture with the model (despite satisfactory meteorological fields) without updated emissions estimates and
knowledge of dynamic local activity data (for instance diurnal activity profiles specific to Indian regions). For example, residential emissions are a major contributor to poor air quality in rural and suburban areas in northern India with an estimated 16 % to 80 % contribution towards SOA components of PM$_{2.5}$ (Rooney et al., 2019).

**Table 7.** Statistical summary of comparisons of monthly mean modelled and observed AOD at 550
nm derived from MODIS and at the 4 AERONET stations from September to November 2016. The statistical metrics are mean bias (MB), normalized mean bias (NMB), mean absolute error (MAE), root mean square error (RMSE) and Pearson's correlation coefficient (*r*).

| Month | MB | NMB | MAE | RMSE | *r* |
|---|---|---|---|---|---|
| **AOD (MODIS)** | | | | | |
| September | 0.25 | 0.69 | 0.28 | 0.34 | 0.72 |
| October | 0.06 | 0.20 | 0.11 | 0.15 | 0.80 |
| November | 0.00 | -0.01 | 0.09 | 0.13 | 0.86 |
| **AERONET** | | | | | |
| **Jaipur** | | | | | |
| September | 0.43 | 0.96 | 0.43 | 0.45 | 0.38 |
| October | 0.10 | 0.31 | 0.14 | 0.17 | 0.04 |
| November | -0.03 | -0.06 | 0.17 | 0.26 | 0.83 |
| **Kanpur** | | | | | |
| September | 0.30 | 0.66 | 0.32 | 0.37 | 0.60 |
| October | -0.01 | -0.02 | 0.19 | 0.25 | 0.64 |
| November | -0.15 | -0.21 | 0.20 | 0.26 | 0.72 |
| **Gandhi College** | | | | | |
| September | 0.02 | 0.04 | 0.22 | 0.27 | -0.08 |
| October | -0.03 | -0.05 | 0.13 | 0.17 | 0.69 |
| **Lahore** | | | | | |
| September | 0.39 | 0.49 | 0.49 | 0.57 | 0.15 |
| October | -0.14 | -0.13 | 0.38 | 0.50 | 0.26 |
| November | -0.37 | -0.37 | 0.44 | 0.65 | 0.75 |




Furthermore, the inaccuracies in simulating individual fractions of total $PM_{2.5}$ also add to the observed model biases; for example, the lack of heterogeneous aqueous phase chemistry in the current aerosol scheme potentially underestimates the aerosol processes involving SOA formation (Tuccella et al., 2012; Balzarini et al., 2015), while high contributions from dust in the MOSAIC scheme could lead to
overestimations (Georgiou et al., 2018). In addition, the earlier studies report large positive biases in simulating surface and column concentrations of trace gases like $NO_x$ (Kumar et al., 2012a) and concentrations of $SO_2$ (Conibear et al., 2018) over urban areas in India. These biases further contribute to uncertainties in simulating reactive trace gas and secondary pollutants.

Significant post-harvest crop residue burning takes place in north-western states of India from late
October to mid-November (Jethva et al., 2019), which impacts the air quality locally as well as in downwind regions of central and eastern IGP (Bhardwaj et al., 2016; Kanawade et al., 2020; Kulkarni et al., 2020; Singh et al., 2021; Mhawish et al., 2022; Govardhan et al., 2023). Other uncertainties in simulating $PM_{2.5}$ concentrations arise from errors in scaling biomass burning emissions estimates which largely depend on the limited number of daily satellite-based retrievals and are sometimes compromised
by dense smoke from fires being misrepresented as cloud cover in the detection algorithm (Cusworth et al., 2018). In their study, Singh et al. (2021) report the annual mean contribution of biomass burning to $PM_{2.5}$ over India to be 8%, but with a strong seasonal dependence (up to 39 % in October-November in Delhi). As previously discussed in the literature, MODIS fire detection is susceptible to missing small fires like agricultural burning (Cusworth et al., 2018; Roozitalab et al., 2021). In addition to the surface
measurements, comparisons with MERRA-2 products highlight a good agreement between the WRF-Chem simulations and the reanalysis approach of employing satellite data assimilations. Navinya et al. (2020) and others, however, find MERRA-2 to underestimate simulated $PM_{2.5}$ over India in comparison to the measurements.

Overall, the evaluation of the WRF-Chem simulated chemistry demonstrates adequate performance
during October and November for $PM_{2.5}$ and is assessed to be suitable to investigate the atmospheric dynamics during extreme pollution events. The modelled results presented here, and in other studies of pollution episodes and aerosol climatology over India, clearly show that the October-November period has higher aerosol loading over most of the domain. A mix of factors like emission patterns, meteorology shifts and topography intensify the existing high pollution levels in some parts of India
(Kulkarni et al., 2020; Kumar et al., 2018a; Mhawish et al., 2022; Kanawade et al., 2020). Sawlani et al. (2019) and Kanawade et al. (2020) attribute the 2016 haze episode to a mix of coinciding factors: local emissions from fireworks, enhanced fire counts from agricultural crop residue burning in northwestern states, stagnant conditions resulting from low temperatures, shallow PBL, weaker northwesterly winds, and high ambient RH. The crop residue burning in 2016 (over Punjab, Haryana
and Uttar Pradesh in northwest India and Pakistan) detected by combined VIIRS and MODIS sensors reveal higher total burning events by up to 30% and 41% compared to 2017 and 2018, respectively



(Chhabra et al., 2019). Similar high pollution events have been reported during post-monsoon months in later years (Dekker et al., 2019; Kulkarni et al., 2020; Takigawa et al., 2020; Roozitalab et al., 2021; Beig et al., 2021; Mhawish et al., 2022). Additionally, a few studies also report a layer of biomass-

burning smoke aerosols at 2-3 km altitude above the IGP region using CALIPSO (Cloud–Aerosol Lidar and Infrared Pathfinder Satellite Observations) retrievals (Shaik et al., 2019; Kumar et al., 2018b), which is detrimental for haze occurrences and modifying local meteorology.

## 5. Conclusions

We comprehensively compare the WRF-Chem v4.2.1 modelled meteorology and aerosol chemistry

with a wide range of observational data that includes ground-based, satellite and reanalysis products over northern India. The simulations are performed at a spatial resolution of 12 km and for the 2016 monsoon (September) to post-monsoon (October-November) transition, with a focus on the severe haze pollution episode from 30 October to 7 November.

The meteorological fields show strong seasonal and spatial variability over the IGP region with a

marked decrease in temperature, WS, and PBLH from monsoon to post-monsoon, most notably for PBLH. Overall, we find that the model accurately represents meteorology during the afternoon hours. The surface daily and diurnal trend in temperature is best reproduced by the model, followed by relative humidity, with negligible biases across all sites. In contrast, daily mean model wind speed is widely biased high (by ~ 0.5 – 0.8 m s$^{-1}$) largely due to strong night-time overestimations (up to 1.5 m s$^{-1}$),

while the afternoon WS is reasonably reproduced by the model. This suggests a potential model failure in surface layer decoupling at night.

Comparison of upper air meteorology with radiosonde profiles shows negligible biases and excellent correlations for temperature and virtual potential temperature ($r > 0.95$) across all sites. The model overestimates wind speed in the lowest layers, consistent with surface observation comparisons whilst

matching well with observed WS aloft. In comparison to MERRA-2 reanalysis products, modelled PBLH generally has negative mean bias of $> 25 \%$ in all the months but agrees well spatially.

Modelled and observed PM$_{2.5}$ concentrations show good agreement (except during September) with overall better correlations for 8 sites averaged across Delhi ($r > 0.6$) and 12 sites across the remaining domain ($r > 0.5$). In September, model concentrations show large biases due to overestimation in dust

generation over the western arid region and possible long-range transport across the measurement sites.

The model simulates the high pollution episode with notable peaks in daily mean PM$_{2.5}$ concentrations but underestimates the exceptionally high observed daily PM$_{2.5}$ (300 – 750 µg m$^{-3}$) by a factor of 2-3. Despite the accurate representation of the vertical temperature gradient, the model underestimates high surface PM$_{2.5}$ concentrations due to stronger simulated WS favouring the dispersion of the surface

pollutants, together with uncertainties in the emissions inventories. Both the model and surface



measurements show that Delhi experiences the highest $PM_{2.5}$ concentrations during the severe pollution episode followed by regional dispersal of pollutants downwind. During the episode, daily simulated anthropogenic $PM_{2.5}$ composition comprised high fractions of nitrate (5 - 25 %) and secondary organic aerosols (10 - 20 %), consistent with previous measurement and modelling studies. The contribution of
BC and primary organic matter to the total simulated $PM_{2.5}$ mass also increases in November.

Comparison with MERRA-2 reanalysis data shows the spatiotemporal distribution of surface $PM_{2.5}$ to have systematic high biases in September along the dry western region of the domain and low bias in October-November in the IGP region. However, the model captures quite well the high $PM_{2.5}$ and BC concentrations over the IGP, including Delhi and upwind biomass burning regions during November.
Variability in modelled AOD compared with satellite retrievals from MODIS is captured very well with $r \geq 0.8$ in October-November. The model likewise compares well with ground-based AERONET measurements of daily AOD ($r \geq 0.69$) across all sites except during September.

Our evaluations consistently reveal the best performance of the model in simulating $PM_{2.5}$ and BC concentrations is for November followed by October, with model underestimations largely stemming
from the extreme episodic nature of the pollution event. The lack of measurement data for individual $PM_{2.5}$ components and the limited spatial coverage of measurement sites restricts the extent of the evaluation of this period. Overall, however, the model is found adequate for subsequent investigation of the vertical distribution of particle components and their interactions with meteorology through sensitivity simulations and improved emissions estimates. Our results also suggest that improved
diurnal characterisation of boundary layer processes could considerably enhance the model performance over this region.

**Code and data availability**

All the data sets used for comparison and source codes for model simulations are openly available. WRF-Chem source code can be obtained from
https://www2.mmm.ucar.edu/wrf/users/download/get_source.html. The ERA-5 input data were downloaded from https://cds.climate.copernicus.eu/cdsapp#!/dataset/reanalysis-era5-pressure-levels?tab=overview. The chemical boundary conditions from MOZART are available at https://www2.acom.ucar.edu/gcm/mozart. All the emissions inputs and pre-processor tools were obtained from https://www2.acom.ucar.edu/wrf-chem/wrf-chem-tools-community. The links for
openly available ground, satellite and reanalysis datasets used for evaluation are provided in Table S2.



**Author contributions**

PA, DSS and MRH conceptualised the study. PA compiled the measurement datasets performed formal
model simulations, and data analyses, curated the data and wrote the text with discussions and
supervision by MRH and DSS. DSS and MRH edited and commented on the text.

**Competing interests**. The authors declare that they have no conflict of interest.

**Acknowledgments**

This work was carried out on the Cirrus UK National Tier-2 HPC Service at EPCC
(http://www.cirrus.ac.uk) funded by the University of Edinburgh and EPSRC (EP/P020267/1). We
acknowledge the WRF-Chem community and the Atmospheric Chemistry Observations and Modeling
Lab (ACOM) of NCAR for providing the preprocessor tools {mozbc, fire_emis, anthro_emis,
bio_emiss} used in this study (https://www2.acom.ucar.edu/wrf-chem/wrf-chem-tools-community last
accessed 29 May 2023). We also acknowledge the use of data provided by GHSL - Global Human
Settlement Layer (https://ghsl.jrc.ec.europa.eu/index.php last accessed 29 May 2023). The geographical
maps were downloaded from https://github.com/IDFCInstitute/IndiaMap_Data (last accessed 29 May
2023). PA is thankful to Alaa Mhawish for providing the MODIS data. The use of open software
packages and python libraries are also gratefully acknowledged.

**Financial support**

PA acknowledges UoE scholarships (Principal's Career Development Scholarships and Edinburgh
Global Research Scholarship). DSS acknowledges support from Natural Environment Research
Council (grant no. NE/S009019/1).

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
