# Peer review of "Evaluation of WRF-Chem simulated meteorology and aerosols over northern India during the severe pollution episode of 2016"

_EGUsphere, 2023_

## Referee Comment (RC2)

**Paper Title:** Evaluation of WRF-Chem simulated meteorology and aerosols over northern India during the severe pollution episode of 2016

**General review**

This paper presents an evaluation of the capabilities of WRF-Chem in replicating seasonal meteorological patterns and aerosol chemistry, with a specific focus on $PM_{2.5}$ and black carbon, across the Indo-Gangetic Plain. The authors have conducted a comparative assessment, comparing the simulations to reanalysis and observational data, in order to assess its performance. The findings of this investigation indicate that the WRF-Chem model is a suitable tool for examining the interplay between aerosols and meteorology during periods of intense pollution. Additionally, the study underscores enhancements in the representation of diurnal boundary layer processes and emission estimations within the model. However, numerous studies have already been conducted to evaluate the performance of WRF-Chem in simulating meteorology and aerosols over the Indian region (Kumar et al., Jena et al., Sengupta et la., etc.). This abundance of existing research makes it challenging to identify the novelty of the current study. Therefore, I recommend a substantial revision of the manuscript and suggest the authors to emphasize on bringing out the novelty of their research before resubmission.

**Major Comments:**

The paper needs a clearer explanation of its scientific motivation. It's crucial to clarify why this analysis is being conducted, especially considering previous publications that have validated WRF-Chem for aerosol studies over the IGP. The authors should provide a strong rationale for their study or highlight the unique aspects that set their work apart from previous research in this area. The authors should include a comparative discussion that highlights how their results, specifically concerning meteorology and aerosol simulation, compare with or differ from existing research. The assessment of aerosol feedback on meteorology needs to be more explicit.

The authors utilized the MOSAIC 4-bin scheme for aerosol chemistry characterization. However, it is unclear whether they incorporated aqueous phase chemistry into their model. Including aqueous phase chemistry is crucial as it

replicates aerosol wet removal processes, especially related to fog/haze formation during winter. These processes significantly impact atmospheric composition and are valuable for air quality research. Unfortunately, this aspect is mostly absent in the current manuscript, with no clear mention of its inclusion. Winter aerosol chemistry in the IGP is notably affected by aqueous phase chemistry, as highlighted in Acharja et al. (2023). The absence of this process in the model introduces uncertainties, a point consistently emphasized throughout the manuscript.

The authors recognize the model's limitations in accurately representing emissions and land use information. It would be helpful to explain the measures taken to mitigate these limitations and clarify how model validation remains meaningful despite these acknowledged challenges.

The authors acknowledge the imperfections in representing emissions but haven't specified the steps taken to mitigate this uncertainty. This study utilized the 2010 EDGAR-HTAPv2.2 emissions dataset to assess air quality from September to November 2016. However, emissions in India have significantly changed over the 6-year period. Using static emissions without accounting for these changes does not offer an accurate evaluation of the model's performance. Simply acknowledging this uncertainty, a point already discussed in previous studies, doesn't add substantial value to this research. Additionally, considering the diurnal cycle in emissions is crucial. The authors should, at the very least, apply the diurnal cycle based on existing literature, rather than omitting it entirely from emissions modeling.

The authors observed that the model overestimates dust in September due to exaggerated wind and underestimated dust deposition. This issue was previously addressed by Kalenderski et al. (2013), who attempted to adjust the model for this discrepancy.

The model generates outputs hourly, and IEM-ASOS weather data for meteorological parameters and CPCB data of PM2.5 are also available at an hourly resolution. However, in the manuscript, model performance metrics are calculated based on monthly averaged modeled versus observed values, and daily mean values are compared in time series plots. This approach does not accurately reflect the model's performance and can be misleading. To assess the actual model

performance, the authors might consider providing performance statistics based on hourly datasets.

In October, there is substantial biomass burning activity in Punjab and Haryana States, impacting air quality in rural and urban areas downwind of the IGP. The FINN emission inventory notably underestimates these fire emissions (Jena et al., 2015). However, the manuscript almost entirely overlooks the discussion and analysis of this significant event.

The claim that "WRF-Chem accurately represents afternoon meteorology and reasonably reproduces wind patterns" needs elaboration. It's crucial to explain how these factors influence the daily fluctuations in PM2.5 and BC concentrations.

The comparison of model-generated PM2.5 and BC concentrations with MERRA-2 global reanalysis data, using the GOCART scheme, raises concerns. Utilizing observational data for validation would enhance the reliability of the results. Reconsidering this approach is advisable for improved credibility.

The modeled PM2.5 composition predominantly consists of nitrate aerosol. However, during winter in Delhi, chloride significantly contributes to a substantial portion of PM2.5 composition (Ali et al., 2019; Pawar et al., 2023). Does your model setup incorporate chloride chemistry, and is chloride emission included in your inventory?

Recommendation: The authors are encouraged to revise the paper to clarify the scientific objectives of their study. It is essential to differentiate their work from existing literature on the topic. To achieve this, they should thoroughly review previous studies and identify gaps in the current state of knowledge. One potential aspect to explore further could be the vertical distribution of aerosols and their intricate interactions with meteorological conditions during peak pollution seasons. By addressing such gaps and specifying their research focus, the authors can revise their paper with a well-defined scientific objective that contributes valuable insights to the existing body of research.

**Minor Comments:**

I have concerns about the model setup. Is there a spin-up period given to the model run, and if so, how long is it? The manuscript lacks this information. The authors mentioned the application of nudging but did not specify whether it is applied in the Planetary Boundary Layer (PBL) or across the entire atmosphere. Additionally, the type of nudging and its method are not clear.

Furthermore, the calculation method for Aerosol Optical Depth (AOD) at 550 nm in WRF-Chem needs clarification. The evaluation statistics of AOD with MODIS data were generated using monthly mean values, while MODIS AOD data are available at daily resolution. To assess true model performance, it is crucial to compare daily mean MODIS AOD with daily modeled AOD.

The manuscript lacks proper scientific justification for the underestimation or overestimation of meteorological parameters, PM2.5, and its composition. This aspect needs to be supported with sound scientific reasoning.

Specifically, in line no. 67, a comma is needed after 'globally.' In line no. 82, the term "End of October" is somewhat vague. It is recommended that the authors specify the exact starting date of the event. Although this information is provided later in the manuscript, including it here would enhance clarity.

---

## Author Comment (AC1)

**Egusphere-2023-1150: Evaluation of WRF-Chem simulated meteorology and aerosols over northern India during the severe pollution episode of 2016**

**Agarwal et al.**

**Responses to Anonymous Reviewer 1**

We thank anonymous reviewer 1 for the time they spent reviewing our manuscript. Our point-by-point responses to the comments are given below in blue.

In this manuscript, Agarwal et al. performed a diagnostic evaluation of the state-of-the-art meteorology chemistry models in their ability to simulate meteorology and air quality over the populous, polluted Indo-Gangetic plain. The simulation is compared to ground and satellite observations as well as reanalysis products. Such an evaluation is useful in that it provides a benchmark for future studies; additionally, the study also shows that more accurate emission inventory and better characterization of boundary layer processes are key for further improvements. The manuscript is overall well-written, and I recommend the publication of the manuscript after minor revisions.

Response: We thank the reviewer for their recommendation to publish, following attention to minor revisions.

Comments:
1. Line 32: The word 'This' refers to insufficient aspects of the modelling, but the previous sentence is stating the model is reasonably good.

Response: We have rephrased the sentence as follows:

"WRF-Chem performs better at simulating the monthly average daytime meteorology. The systematically overpredicted wind velocities (more prominent during the night) lead to enhanced dilution and mixing, which, combined with underestimated input emissions, are responsible for pollutant underestimations in the post-monsoon season."

2. Line 35: The author should simply point out that better diurnal characterization of the boundary layer processes and emission estimates are necessary. Whether or not such improvement makes the model suitable to understand aerosol feedbacks on meteorology remains to be demonstrated, considering that 'feedbacks' is not really discussed in the current study.

Response: We acknowledge this point and have rephrased the sentence to read:

"Overall, the model realistically captures the seasonal and spatial gradient of meteorology and ambient pollutants over northern India and highlights the need for improved emissions estimates for a better representation of complex aerosol chemistry during extreme episodic pollution."

3. Line 164: Can the author say a bit more than 'satisfactorily' so that the reader better understands the bias from the inventory?

Response: We have revised our text to point out biases reported for the EDGAR inventory:

"Moreover, compared with other global inventories of coarser resolution (e.g., ECLIPSE), the use of EDGAR-HTAPv2.2 has been found to simulate air quality over India with a greater local heterogeneity and to show slightly smaller overall biases when compared to reanalysis and satellite products (Upadhyay et al., 2020)."

4.  Line 295: What is the reason for the better correlation in September? Is it due to the monsoon season?

Response: A better correlation for September is likely due to seasonal differences in that a generally higher WS in this month is also associated with dominant westerly and southerly wind flow compared to October and November. The model can better predict the higher wind speeds prevalent during September compared with the low wind speeds during October and November.

5.  Line 317: ERA-5 has positive bias in RH and is used to drive WRF-CHEM, how does it propagate to the negative bias in the simulations?

Response: We note from previous studies that systematic underestimations in simulating RH by WRF-Chem are reported when using other global datasets to drive the meteorological initial and boundary conditions (Ansari et al., 2019; Gunwani et al., 2023). So, the underestimation of RH in WRF-Chem likely stems from, first, the warm bias in the model and, second, the tendency to drift away from the large-scale driving prescribed meteorology towards drier conditions (Jain and Kar, 2017) because of uncertainties in closure assumptions in the convective parameterisation for numerical weather prediction (Grell and Dévényi, 2002). Additionally, the fact that the nudging coefficient is small, which allows for the model to simulate its own dynamic meteorology, and that no nudging is applied to meteorological variables in the planetary boundary layer may also explain why the model is still overall negatively biased for RH compared to ERA5.

6.  Figure 1: The font size of the ticklabels should be increased. Fig 1a. Are there any water grid cells?

Response: We have increased the tick labels as shown in the revised Figure 1 below. There are very few water grid cells in the domain.

[Figure]

7. Figure 4 is wrongly placed before Figure 3. Also, the caption of Figure 3 is incomplete.

Response: We apologise for these two formatting errors in our Discussion paper which are now rectified in the revised MS.

8. Figures 6&7: The size of panels with similar contents should be adjusted to have the same size.

Response: We have made the panel size adjustments in the revised MS. Please find the revised figures below:

[Figure]

**Figure 6.** Time series of daily means (left) and mean diurnal cycles (right) of modelled individual PM$_{2.5}$ components averaged across 8 stations in Delhi and 12 stations over the rest of the domain (labelled 'Others') from September – November 2016. The individual species contribution abbreviations are: SOA (secondary organic aerosol), POA (primary organic aerosol), SO$_4^{2-}$ (sulfate), NH$_4^+$ (ammonium), NO$_3^-$ (nitrate), BC (black carbon). The vertical dashed lines delineate the period of severe high pollution between 30 October and 7 November. Note the different x-axis scales on each side.

[Figure]

**Figure 7.** Spatial distributions of monthly mean concentrations (µg m⁻³) of a) PM₂.₅ and b) black carbon from the WRF-Chem model and MERRA-2 for September to November 2016. The monthly mean PM₂.₅ at the measurement sites are shown in circles in a).

9.  Fig S2: I find the colors for ERA and MERRA very difficult to distinguish.

Response: We believe the reviewer is referring to Fig. S3 and have adjusted the colour scheme.

[Figure]

**Figure S3.** Scatter plots of daily mean measured and modelled surface meteorology variables derived from ERA-5, MERRA-2 and WRF-Chem across the 49 ASOS measurement sites for each of the 3 months of the study period: 2 m temperature, relative humidity (%), and wind speed (m s[-1]). The 1:1 line is shown as red dashed.

Technical:
10. Line: 285: 'very slightly low' -> slightly lower

Response: The sentence has been amended as suggested

11. Line 572: revise the sentence 'including because...'

The sentence has been amended as below:

"In addition, nearly all the measurement sites are in or near dense urban areas with heavy influences from traffic and local anthropogenic activities (for example, trash burning and residential cooking), which are not reflected in the monthly anthropogenic emission inputs."

References cited in responses to Reviewer 1

Ansari, T. U., Wild, O., Li, J., Yang, T., Xu, W., Sun, Y., and Wang, Z.: Effectiveness of short-term air quality emission controls: a high-resolution model study of Beijing during the Asia-Pacific Economic Cooperation (APEC) summit period, Atmos. Chem. Phys., 19, 8651–8668, https://doi.org/10.5194/acp-19-8651-2019, 2019.

Grell, G. A. and Dévényi, D.: A generalized approach to parameterizing convection combining ensemble and data assimilation techniques, Geophysical Research Letters, 29, 38-1-38–4, https://doi.org/10.1029/2002GL015311, 2002.

Gunwani, P., Govardhan, G., Jena, C., Yadav, P., Kulkarni, S., Debnath, S., Pawar, P. V., Khare, M., Kaginalkar, A., Kumar, R., Wagh, S., Chate, D., and Ghude, S. D.: Sensitivity of WRF/Chem simulated PM2.5 to initial/boundary conditions and planetary boundary layer parameterization schemes over the Indo-Gangetic Plain, Environ Monit Assess, 195, 560, https://doi.org/10.1007/s10661-023-10987-3, 2023.

Jain, S. and Kar, S. C.: Transport of water vapour over the Tibetan Plateau as inferred from the model simulations, Journal of Atmospheric and Solar-Terrestrial Physics, 161, 64–75, https://doi.org/10.1016/j.jastp.2017.06.016, 2017.

---

## Author Comment (AC2)

**egusphere-2023-1150: Evaluation of WRF-Chem simulated meteorology and aerosols over northern India during the severe pollution episode of 2016**
**Agarwal et al.**

**Responses to Anonymous Reviewer 2**

We thank anonymous Reviewer 2 for their time spent reviewing our manuscript and the suggested recommendations. Our point-by-point responses to the comments and our revisions in the manuscript are given below in blue.

This paper presents an evaluation of the capabilities of WRF-Chem in replicating seasonal meteorological patterns and aerosol chemistry, with a specific focus on PM2.5 and black carbon, across the Indo-Gangetic Plain. The authors have conducted a comparative assessment, comparing the simulations to reanalysis and observational data, in order to assess its performance. The findings of this investigation indicate that the WRF-Chem model is a suitable tool for examining the interplay between aerosols and meteorology during periods of intense pollution. Additionally, the study underscores enhancements in the representation of diurnal boundary layer processes and emission estimations within the model. However, numerous studies have already been conducted to evaluate the performance of WRF-Chem in simulating meteorology and aerosols over the Indian region (Kumar et al., Jena et al.,Sengupta et la., etc.). This abundance of existing research makes it challenging to identify the novelty of the current study. Therefore, I recommend a substantial revision of the manuscript and suggest the authors to emphasize on bringing out the novelty of their research before resubmission.

1. The paper needs a clearer explanation of its scientific motivation. It's crucial to clarify why this analysis is being conducted, especially considering previous publications that have validated WRF-Chem for aerosol studies over the IGP. The authors should provide a strong rationale for their study or highlight the unique aspects that set their work apart from previous research in this area. The authors should include a comparative discussion that highlights how their results, specifically concerning meteorology and aerosol simulation, compare with or differ from existing research. The assessment of aerosol feedback on meteorology needs to be more explicit.

Response: We have enhanced the motivation for our work and the reference to prior work in several places in our paper. Here, we have added additional discourse on the performance of our WRF-Chem modelling study over India compared to previous literature. Regarding this, we revise the introduction to bring more clarity to the objective and novelty of our study as provided below:

"This study aims to evaluate the WRF-Chem regional atmospheric chemistry transport model's ability to simulate the spatiotemporal seasonal meteorology and aerosol chemistry across north India and the IGP in September-November 2016. Our choice to analyse the 2016 seasonality and the pollution episode differs from previous literature (Kumar et al., 2020; Jena et al., 2020; Sengupta et al., 2022; Govardhan et al., 2023) in several aspects:

(i) We use an updated WRF-Chem version (v4.2.1) and utilise the MOZART-MOSAIC chemical scheme (detailed in Section 2.1), which explicitly represents the chemistry of secondary organic and inorganic aerosols that make up the dominant components of $PM_{2.5}$ in the post-monsoon season, as compared to the less detailed bulk GOCART scheme used in these earlier studies.

(ii) The 2016 pollution episode over the IGP was one of the worst for air quality (since 2004) and anomalous for the highest rice crop production (since 2002) in NW Indian states, resulting in high crop residue burning in that year (Voiland and Jethva, 2017; Jethva et al., 2019; Sembhi et al., 2020). As shown by multiple trend analyses, 2016 had the highest number of agricultural fires of the last decade or so (Sarkar et al., 2018; Mukherjee et al., 2018; Thomas et al., 2019; Kulkarni et al., 2020; Sembhi et al., 2020; Liu et al., 2021; Jethva, 2022; Gupta et al., 2023). (A time series of annual fire counts (Figure 1) in four northern Indian states is shown below to illustrate this point here, although it is not necessary to include this figure in the revised paper itself given the substantial citations we provide in the revised text for this fact.) Moreover, although several modelling studies have analysed the air quality during intense post-monsoon episodes in the years after 2016 (Dekker et al., 2019; Beig et al., 2019; Kulkarni et al., 2020; Roozitalab et al., 2021), studies for 2016 are fewer (Sembhi et al., 2020; Mukherjee et al., 2020). It is, therefore, necessary to understand the implications of this particularly extreme episode with a chemistry transport model whose performance at simulating prevailing seasonal meteorology over a sufficiently long period has been evaluated.

(iii) The use of ERA5 reanalysis data to drive the model meteorology and a comprehensive comparison of the simulated meteorology and biases across northern India is an additional novelty of this work, which has not been documented so far. With this in mind, this study aims to improve our understanding of the spatiotemporal meteorology and chemistry across northern India and the IGP in September-November 2016, starting with reporting the performance of WRF-Chem using improved chemical schemes and input meteorological boundary conditions. A further novelty is our use of a wide range of ground and satellite observations as well as reanalysis products in the evaluation."

The assessment of aerosol feedback on meteorology is not the focus of the current study, which is the detailed evaluation of the model. We apologise for the inadvertent mention of this in the original paper, and it has now been removed from the revised manuscript (MS).

[Figure]

Figure 1. S-NPP VIIRS observed total fire counts in four Indian states during the post-monsoon (September 6 to November 30) season over a period of 9 years. (Adapted from Gupta et al., 2023, Figure 3.)

2. The authors utilized the MOSAIC 4-bin scheme for aerosol chemistry characterization. However, it is unclear whether they incorporated aqueous phase chemistry into their model. Including aqueous phase chemistry is crucial as it replicates aerosol wet removal processes, especially related to fog/haze formation during winter. These processes significantly impact atmospheric composition and are valuable for air quality research. Unfortunately, this aspect is mostly absent in the current manuscript, with no clear mention of its inclusion. Winter aerosol chemistry in the IGP is notably affected by aqueous phase chemistry, as highlighted in Acharja et al. (2023). The absence of this process in the model introduces uncertainties, a point consistently emphasized throughout the manuscript.

Response: First, we clarify that the MOSAIC-MOZART 4-bin chemistry mechanism chosen for the current study doesn't include the detailed aqueous-phase reactions such as those described in the recently published article Acharja et al. (2023). The model does include aerosol wet removal processes.

Aqueous-phase reactions play an important role in aerosol chemistry and particle size during winter fog formation over northern India when the associated aerosol water content is also high (Bharali et al., 2019, 2023). Our study covers the period September-November, during which the average relative humidity (RH) across the main region of interest, the IGP, is not as high as reported during the winter (December – February). Therefore, we expect that the uncertainties due to the lack of detailed aqueous-phase chemistry, which is highly dependent on the RH, are likely small in comparison to the other limitations in emissions and boundary layer dynamics during the study period. When we discuss uncertainties, we acknowledge that the lack of detailed aqueous-phase chemistry may be one of the model's limitations.

The MOSAIC 4-bin chemistry mechanism used here does include detailed aerosol solid, liquid and mixed-phase equilibria and thermodynamic gas-particle partitioning to compute aerosol composition and a simple parameterisation of SOA aqueous chemistry using glyoxal (Knote et al., 2014), but does not explicitly include the detailed aqueous-phase chemistry, as described for example in Acharja et al. (2023). The aerosol processes in the mechanism we have used include aerosol transport, dry and wet removal, water uptake, nucleation, coagulation and condensation processes (more details can be found in Jan Kazil, 2015). Neutralisation/condensation of sulfuric and nitric acid to form ammonium sulfate, ammonium bisulfate and ammonium nitrate are controlled by solid-liquid thermodynamic equilibria. These mechanisms are solved by MESA (Multicomponent Equilibrium Solver for Aerosols) and MTEM (Multicomponent Taylor Expansion Method) (Zaveri et al., 2008) modules that calculate water content in each size bin and interparticle solid-liquid or mixed phase equilibria in multicomponent aqueous aerosols. The scheme also considers spontaneous uptake of water by dry aerosol particles as a function of RH using the mutual deliquescence RH (MDRH) theory for multicomponent aerosols, allowing for a more realistic representation of secondary aerosol formation in highly polluted environments as also considered by Acharja et al. (2023). So, whilst a detailed aqueous phase chemistry is not included, many important aqueous phase aerosol processes are simulated in our mechanism.

To reflect the reviewer's comment, we have modified the text in **Sections 2.1 and 4.5** to provide more clarity and detail to our model set-up and the resulting implications on the model simulations in discussions. Additionally, the lack of phase partitioning of HCl gas from the available chemistry suite in WRF-Chem is also acknowledged (see our response to comment number 10 below) as provided below:

"Aerosol chemistry is simulated using the Model for Simulating Aerosol Interactions and Chemistry (MOSAIC) 4-bin scheme (Zaveri et al., 2008). MOSAIC mechanism includes detailed solid, liquid and mixed-phase equilibria and thermodynamic gas-particle partitioning to compute aerosol composition and a simple parameterisation of SOA aqueous chemistry using glyoxal (Knote et al., 2014), but does not explicitly include detailed aqueous-phase chemistry, such as that described in Acharja et al. (2023). The aerosol processes in the mechanism include aerosol transport, dry and wet removal, water uptake, nucleation, coagulation, and condensation processes."

"The lack of aqueous-phase chemistry in our model framework further adds some biases in reproducing accurate amounts of secondary aerosol components of PM ($SO_4^{2-}$, $NH_4^+$, $NO_3^-$) and their subsequent scavenging by aqueous chemistry in the cloud or water droplets."

3. The authors recognize the model's limitations in accurately representing emissions and land use information. It would be helpful to explain the measures taken to mitigate these limitations and clarify how model validation remains meaningful despite these acknowledged challenges.

Response: As also described in our response to the reviewer's comment 4 below, in future work, we will improve our model set-up by moving to anthropogenic and fire emissions datasets that were not available to us at the time of the present work. While updated land-use and land-cover information is important for regional and local meteorology, we must currently rely on WRF-compatible data made available by the NCAR. We discuss the shortcomings in emissions data in our paper (see next response), but we do not think the emissions and land-use datasets we have used make our model evaluation not meaningful.

4. The authors acknowledge the imperfections in representing emissions but haven't specified the steps taken to mitigate this uncertainty. This study utilized the 2010 EDGAR-HTAPv2.2 emissions dataset to assess air quality from September to November 2016. However, emissions in India have significantly changed over the 6-year period. Using static emissions without accounting for these changes does not offer an accurate evaluation of the model's performance. Simply acknowledging this uncertainty, a point already discussed in previous studies, doesn't add substantial value to this research. Additionally, considering the diurnal cycle in emissions is crucial. The authors should, at the very least, apply the diurnal cycle based on existing literature, rather than omitting it entirely from emissions modelling.

Response: First, we clarify that the information about diurnal variation applied to the anthropogenic emissions in our configuration is provided in our original paper (L160 – 161). To reiterate this, we did apply a simple diurnal profile (day/night) to the input anthropogenic emissions following the WRF-Chem anthropogenic emission preprocessor tool (details can be found at https://www2.acom.ucar.edu/wrf-chem/wrf-chem-tools-community, last accessed 16 Nov 2023). Secondly, we have now added more information regarding the implications of using the 2010 anthropogenic emissions inventory on the results of the present work. We will update our model to incorporate the EDGARv5 (Crippa et al., 2021) and FINNv2.5 (Wiedinmyer et al., 2023) emissions as part of our future sensitivity analysis work. We also note that EDGAR-HTAPv2.2 based on 2010 emissions is a common and inevitable source of uncertainty found in many other WRF-Chem modelling studies simulating air quality in the Indian region but has been found to be overall

appropriate (Sharma et al., 2017; Conibear et al., 2018; Roozitalab et al., 2021; Jat et al., 2021; Mogno et al., 2021; Nagar and Sharma, 2022; Pawar et al., 2023).

Following this, we revised our text in **Section 2.1** as shown below:

" Emissions over India evolved from 2010 to 2016 (the year of observations we evaluate against). This mismatch adds some uncertainty to our simulations. Emissions of OC, CO, $NO_X$, $SO_2$, and NMVOC from anthropogenic sectors such as industrial and energy sectors increased because of rapidly increasing demands, whilst primary particulate emissions of BC, OC and $PM_{2.5}$ from residential and informal industry sectors reduced due to cleaner fuel policies (such as the Ujjawala scheme; http://www.pmujjwalayojana.in/) (McDuffie et al., 2020). The decadal estimates based on the global CEDS inventory reported by McDuffie et al. (2020) show an increase from 2010 to 2020 in annual $NH_3$, CO, $SO_2$, $NO_X$, and NMVOC emissions over India from road transport, energy, industry and agricultural sectors. These changes in emissions may mean our model simulations underestimate the BC, primary OC, and secondary aerosol contributions to total PM. However, it is challenging to isolate the impact of these changes in an atmospheric chemistry model because the model output also depends substantially on, among other things, the meteorology, and online emissions. Moreover, compared with other global inventories of coarser resolution (e.g., ECLIPSE), the use of EDGAR-HTAPv2.2 has been found to simulate air quality over India with a greater local heterogeneity and to show slightly smaller overall biases when compared to reanalysis and satellite products (Upadhyay et al., 2020)."

5. The authors observed that the model overestimates dust in September due to exaggerated wind and underestimated dust deposition. This issue was previously addressed by Kalenderski et al. (2013), who attempted to adjust the model for this discrepancy.

Response: We thank the reviewer for drawing our attention to this. We have added the similar observation of Kalenderski et al. to our revised paper and discussed its implications on the results. We will consider similar adjustments in our future work with the model. The following is the updated text in **Section 4.5**.

"Overestimated modelled dust aerosols were also observed by Kalenderski et al. (2013) and Zhao et al. (2010), who tuned the online dust emission flux calculation based on region-specific AERONET measurements for a dust event and found the modelled AOD estimates to improve. "

6. The model generates outputs hourly, and IEM-ASOS weather data for meteorological parameters and CPCB data of PM2.5 are also available at an hourly resolution. However, in the manuscript, model performance metrics are calculated based on monthly averaged modeled versus observed values, and daily mean values are compared in time series plots. This approach does not accurately reflect the model's performance and can be misleading. To assess the actual model performance, the authors might consider providing performance statistics based on hourly datasets.

Response: We point out that the model-observation comparisons are indeed undertaken at native hourly resolution of IEM-ASOS weather data and for the $PM_{2.5}$ and AOD datasets. We have amended the description of the tables reporting the model-observation metrics, which may have led to the confusion.

7. In October, there is substantial biomass burning activity in Punjab and Haryana States, impacting air quality in rural and urban areas downwind of the IGP. The FINN emission inventory notably underestimates these fire emissions (Jena et al., 2015). However, the manuscript almost entirely overlooks the discussion and analysis of this significant event.

Response: We note that L604–L615 (of the original paper) discusses this very point, which is quoted below.

"Significant post-harvest crop residue burning takes place in north-western states of India from late October to mid-November (Jethva et al., 2019), which impacts the air quality locally as well as in downwind regions of the central and eastern IGP (Bhardwaj et al., 2016; Kanawade et al., 2020; Kulkarniet al., 2020; Singh et al., 2021; Mhawish et al., 2022; Govardhan et al., 2023). Other uncertainties in simulating $PM_{2.5}$ concentrations arise from errors in scaling biomass burning emissions estimates, which largely depend on the limited number of daily satellite-based retrievals and are sometimes compromised by dense smoke from fires being misrepresented as cloud cover in the detection algorithm (Cusworth et al., 2018). In their study, Singh et al. (2021) report the annual mean contribution of biomass burning to $PM_{2.5}$ over India to be 8%, but with a strong seasonal dependence (up to 39 % in October-November in Delhi). As previously discussed in the literature, MODIS fire detection is susceptible to missing small fires like agricultural burning (Cusworth et al., 2018; Roozitalab et al., 2021). "

As we note above, in future work we will update to recently released FINNv2.5 fire emissions data that were not available at the time of the present work.

8. The claim that "WRF-Chem accurately represents afternoon meteorology and reasonably reproduces wind patterns" needs elaboration. It's crucial to explain how these factors influence the daily fluctuations in PM2.5 and BC concentrations.

Response: That is a good point. In the original paper, in lines L330-L335 and in L409-L416 corresponding to **Sections 3.1 and 4.1**, we note that WRF-Chem reasonably predicts the diurnal cycle of meteorology compared to measurements, but where there are discrepancies, this will influence diurnal cycles in the pollutant concentrations. Here, we also quote these lines below:

"The associated turbulent fluxes and thermodynamic exchanges occurring in the atmospheric boundary layer are important for model simulated PBL and pollutant dispersal (Shen et al., 2023; Nelli et al., 2020). However, during the extreme pollution episode (30 October to 7 November) both model and observations agree on a reduction in WS (although with varying magnitudes) and a shift in WD. These changes highlight the role of stagnant meteorology in greatly enhancing the near-surface pollution lasting over a week."

"The spatially averaged diurnal cycle for modelled surface $PM_{2.5}$ shows a pronounced diurnal trend matching observations for Delhi sites, while the diurnal cycle is less pronounced at Others sites.

Generally, diurnal trends are in good agreement across all sites, although on average the model tends to underpredict the afternoon dips and night-time peaks compared to the observations, indicating missing anthropogenic activities from the simplified diurnal emissions patterns derived from monthly estimates used in the model. The lack of a representation of a realistic diurnal activity cycle in the anthropogenic emissions highlights meteorology could be driving the modelled $PM_{2.5}$ variation. Although this might partly be affected by the imperfectly represented diurnal variability of WS in the model (Section 3.1)."

9. The comparison of model-generated PM2.5 and BC concentrations with MERRA-2 global reanalysis data, using the GOCART scheme, raises concerns. Utilizing observational data for validation would enhance the reliability of the results. Reconsidering this approach is advisable for improved credibility.

Response: We agree with and appreciate the reviewer's concerns, and we clarify that the evaluation of $PM_{2.5}$ is indeed based on the openly accessible observational data, while the BC measurements are not openly available. This is reported in Section 2.3 of the model setup, discussed in detail in results Section 4.1, and shown in Figure 5 of the original paper. However, due to the limited spatial coverage of $PM_{2.5}$ monitoring sites (only 12 in the entire domain, outside of Delhi), we also compared our results with global MERRA-2 reanalysis aerosol data. We acknowledge the utilisation of the global reanalysis that uses a simpler GOCART scheme for aerosols is not as helpful as the measurements. In our paper, we note the limitation of the MERRA-2 chemical scheme, which lacks treatment of nitrate, ammonium and secondary organic aerosols constituting a substantial portion of fine PM across northern India. MERRA-2 reanalysis assimilates the latest satellite observation-based data to provide aerosol outputs. Comparing with MERRA-2 is therefore, a convenient way to include satellite-derived observations in our evaluation. Therefore, we believe that the comparison serves as an instructive method to understand the extent to which a detailed size-resolved chemical scheme in the WRF-Chem model compares against a relatively simpler aerosol scheme but with assimilated satellite-derived aerosol observations. Additionally, in our original paper, we have kept the dominant part of our evaluation content focused on the observation-model comparisons. We further emphasise these limitations in the revised MS as follows and clarify that the point about the assimilation of satellite data in MERRA-2 is mentioned in the original text in lines L241 – 245 (also quoted below):

"The aerosols in the GOCART scheme are externally mixed and exclude the treatment of nitrate, ammonium, and secondary organic aerosols (Randles et al., 2017). This adds uncertainty to the $PM_{2.5}$ concentration comparisons between WRF-Chem and MERRA-2. However, it still serves as a useful inter-comparative assessment of spatial and seasonal trend of aerosols between the two model-derived datasets. AOD in MERRA-2 is assimilated using multiple satellite and ground-based observation data, including bias-corrected AOD from the Moderate Resolution Imaging Spectroradiometer (MODIS), Advanced Very High-Resolution Radiometer (AVHRR) instruments, Multi-angle Imaging Spectroradiometer (MISR) and Aerosol Robotic Network (AERONET). The aerosol assimilation uses satellite radiance and albedo from observing sensors and bias-corrected AOD, described in detail in (Randles et al., 2017)."

10. The modeled PM2.5 composition predominantly consists of nitrate aerosol. However, during winter in Delhi, chloride significantly contributes to a substantial

portion of PM2.5 composition (Ali et al., 2019; Pawar et al., 2023). Does your model setup incorporate chloride chemistry, and is chloride emission included in your inventory?

Response: We thank the reviewer for this point. Chloride chemistry is not available in the WRF-Chem MOSAIC suite used here. The lack of chloride may certainly contribute to the $PM_{2.5}$ underestimation. We have now added the following text in **Section 4.5**:

"Furthermore, underestimations in modelled $PM_{2.5}$ concentrations across Delhi could also be due to the lack of input emissions of hydrogen chloride (HCl) gas, typically from local rubbish and crop residue burning, which adds substantial chloride aerosols to total $PM_{2.5}$ here by its partitioning between gas and aerosol phases (Cash et al., 2021; Lalchandani et al., 2022; Pawar et al., 2023)."

11. Recommendation: The authors are encouraged to revise the paper to clarify the scientific objectives of their study. It is essential to differentiate their work from existing literature on the topic. To achieve this, they should thoroughly review previous studies and identify gaps in the current state of knowledge. One potential aspect to explore further could be the vertical distribution of aerosols and their intricate interactions with meteorological conditions during peak pollution seasons. By addressing such gaps and specifying their research focus, the authors can revise their paper with a well-defined scientific objective that contributes valuable insights to the existing body of research.

Response: We thank the reviewer for these helpful remarks. Consequently, we have made a number of considerable improvements to the introduction, methods description and discussion of results in our revised paper, as we have detailed in our point-by-point responses above. We will consider the vertical distributions of aerosols in our future work; we considered adding that information here but decided it was beyond the scope of this paper.

**Minor Comments:**

12. I have concerns about the model setup. Is there a spin-up period given to the model run, and if so, how long is it? The manuscript lacks this information. The authors mentioned the application of nudging but did not specify whether it is applied in the Planetary Boundary Layer (PBL) or across the entire atmosphere. Additionally, the type of nudging and its method are not clear.

Response: We have added information about model spin up (in **Section 2.1**) and that there was no nudging of meteorological variables in the PBL, at relevant places in the revised text as provided below. Grid analysis nudging is implemented here which uses point by point relaxation terms in every grid cell towards the spatially and temporally interpolated reanalysis data. We revise and incorporate this detail in our revised MS as follows:

"The WRF model temperature, winds and water vapour are nudged at every 6-hour interval, using grid-nudging and nudging coefficient of $6 \times 10^{-4}$ $s^{-1}$ to ERA5 values with no nudging of these variables within the PBL layer (Stauffer and Seaman, 1994)."

"For our evaluation of WRF-Chem performance, hourly simulations are conducted for 01 September to 30 November 2016, allowing six days of spin-up (from 25 – 30 August)."

13. Furthermore, the calculation method for Aerosol Optical Depth (AOD) at 550 nm in WRFChem needs clarification. The evaluation statistics of AOD with MODIS data were generated using monthly mean values, while MODIS AOD data are available at daily resolution. To assess true model performance, it is crucial to compare daily mean MODIS AOD with daily modeled AOD

Response: This is mentioned in L260 of the original text (quoted below) with the reference where further details on the calculation can be found. We also confirm that AOD comparisons are based on hourly data from MODIS and model.

"AOD in WRF-Chem is simulated between wavelengths 300 - 1000 nm and interpolated to 550 nm using the Ångström power law (Ångström, 1964; Kumar et al., 2014)."

14. The manuscript lacks proper scientific justification for the underestimation or overestimation of meteorological parameters, PM2.5, and its composition. This aspect needs to be supported with sound scientific reasoning.

Response: This comment is a summary of points already made by the reviewer, to which we have responded individually above.

15. Specifically, in line no. 67, a comma is needed after 'globally.' In line no. 82, the term "End of October" is somewhat vague. It is recommended that the authors specify the exact starting date of the event. Although this information is provided later in the manuscript, including it here would enhance clarity

Response: The requested amendments have been made.

References cited in responses to Reviewer 2

Acharja, P., Ghude, S. D., Sinha, B., Barth, M., Govardhan, G., Kulkarni, R., Sinha, V., Kumar, R., Ali, K., Gultepe, I., Petit, J.-E., and Rajeevan, M. N.: Thermodynamical framework for effective mitigation of high aerosol loading in the Indo-Gangetic Plain during winter, Sci Rep, 13, 13667, https://doi.org/10.1038/s41598-023-40657-w, 2023.

Ångström, A.: The parameters of atmospheric turbidity, Tellus, 16, 64–75, https://doi.org/10.3402/tellusa.v16i1.8885, 1964.

Beig, G., Srinivas, R., Parkhi, N. S., Carmichael, G. R., Singh, S., Sahu, S. K., Rathod, A., and Maji, S.: Anatomy of the winter 2017 air quality emergency in Delhi, Science of The Total Environment, 681, 305–311, https://doi.org/10.1016/j.scitotenv.2019.04.347, 2019.

Bharali, C., Nair, V. S., Chutia, L., and Babu, S. S.: Modeling of the Effects of Wintertime Aerosols on Boundary Layer Properties Over the Indo Gangetic Plain, Journal of Geophysical Research: Atmospheres, 124, 4141–4157, https://doi.org/10.1029/2018JD029758, 2019.

Bharali, C., Barth, M., Kumar, R., Ghude, S. D., Sinha, V., and Sinha, B.: Role of atmospheric aerosols in severe winter fog over Indo Gangetic Plains of India: a case study, Aerosols/Atmospheric Modelling

and Data Analysis/Troposphere/Physics (physical properties and processes), https://doi.org/10.5194/egusphere-2023-1686, 2023.

Cash, J. M., Langford, B., Di Marco, C., Mullinger, N. J., Allan, J., Reyes-Villegas, E., Joshi, R., Heal, M. R., Acton, W. J. F., Hewitt, C. N., Misztal, P. K., Drysdale, W., Mandal, T. K., Shivani, Gadi, R., Gurjar, B. R., and Nemitz, E.: Seasonal analysis of submicron aerosol in Old Delhi using high-resolution aerosol mass spectrometry: chemical characterisation, source apportionment and new marker identification, Atmospheric Chemistry and Physics, 21, 10133–10158, https://doi.org/10.5194/acp-21-10133-2021, 2021.

Conibear, L., Butt, E. W., Knote, C., Arnold, S. R., and Spracklen, D. V.: Stringent Emission Control Policies Can Provide Large Improvements in Air Quality and Public Health in India, GeoHealth, 2, 196–211, https://doi.org/10.1029/2018GH000139, 2018.

[revised manuscript text omitted]